# Modelling Microplastic Dynamics in Estuaries: A Comprehensive Review, Challenges and Recommendations

Betty John Kaimathuruthy<sup>1</sup>, Isabel Jalón-Rojas<sup>1</sup>, and Damien Sous<sup>2</sup>

<sup>1</sup>Univ. Bordeaux, CNRS, Bordeaux INP, EPOC, UMR 5805, 33600 Pessac, France

<sup>2</sup>Université de Pau et des Pays de l'Adour, E2S UPPA, SIAME, Anglet, France

Correspondence: Isabel Jalón-Rojas (isabel.jalon-rojas@u-bordeaux.fr)

#### Abstract.

The study of microplastic transport and fate in estuaries poses significant challenges due to the complex, dynamic nature of these ecosystems and the diverse characteristics of microplastics. Process-based numerical models have become indispensable for studying microplastics, complementing observational data by offering insights into transport processes and dispersion trends that are difficult to capture through in-situ measurements alone. Effective model implementations require an accurate representation of the hydrodynamic conditions, relevant transport processes, particle properties, and their dynamic behaviour and interactions with other environmental components. In this paper, we provide a comprehensive review of the different process-based modelling approaches used to study the transport of microplastics in estuaries, including Eulerian analytical 2DV models, Eulerian Numerical Models, Lagrangian Numerical Models, and Population Balance Equation Models. We detail each approach and analyze previous applications, examining key aspects such as parameterizations, input data, model setups, and validation methods. We assess the strengths and limitations of each approach and provide recommendations, good practices, and future directions to address challenges, improve the accuracy of predictions, and advance modelling strategies, ultimately benefiting the research field.

# 1 Introduction

The influx of plastic into aquatic systems has reached alarming levels in recent years. The study by Jambeck et al. (2015) estimated that 4 to 12 million metric tonnes of plastic waste enter the ocean annually, with continuing research by Borrelle et al. (2020) indicating this range increases to as much as 23 million metric tonnes. As a result of inadequate waste management practices, plastic waste can quickly enter various pathways into marine habitats. These aquatic ecosystems harbour both macroplastics (larger than 5mm) and microplastics (ranging from 0.1 μm to 5 mm), posing significant threats to organisms, biodiversity, and socio-economic activities (Thompson et al., 2004; Naidu et al., 2018; Beaumont et al., 2019; Hu et al., 2019; Gola et al., 2021).

Estuaries, the dynamic ecosystems where rivers and oceans intertwine, can serve as primary pathways of plastics at the land-sea interface. Depending on the prevailing hydrodynamic conditions, estuaries can act as both sources and sinks of matter, including plastics (Defontaine et al., 2020; Díez-Minguito et al., 2020; López et al., 2021; Biltcliff-Ward et al., 2022; Shi et al.,

2022). This dual role, combined with the high ecological value, underscores the importance of studying plastic transport and dynamics in estuarine environments. However, such studies are quite challenging for several reasons. Firstly, estuaries are subject to complex hydrodynamics, including substantial tidal ranges, dynamic currents, and turbulence (Uncles, 2002; Wollast, 2003; Ganju et al., 2016). Estuaries exhibit spatial and temporal variability of water flows, salinity, temperature, and sediment concentrations (Jalón-Rojas et al., 2015; MacCready et al., 2018), all potentially influencing transport processes of plastic particles that range from small-scale turbulence-driven mixing to large-scale ocean circulation (Fig. 1), resulting in complex dispersion patterns. Nunez et al. (2021) identified the significance of tides and estuarine morphology on the accumulation of plastic debris. Readers seeking a state-of-the-art description of microplastic transport processes in estuaries can refer to Jalón-Rojas et al. (2024a). Secondly, plastics can reach the estuaries through several sources such as rivers, coastal areas, and directly from adjacent run-off, wastewater treatment plants or industrial discharge (Conley et al., 2019; López et al., 2021; Gupta et al., 2021). These multi-entry points also complicate tracing the origin and tracking the transport of plastics. Furthermore, plastic particles come in various sizes, shapes, densities and compositions. These characteristics influence the behaviour of plastics in water, including buoyancy, vertical velocity and aggregation potential (Khatmullina and Isachenko, 2017; Waldschlager and Schuttrumpf, 2019a). Understanding plastic dynamics requires considering the particle-specific properties and physicochemical processes (Fig. 1). In sedimentary environments such as estuaries, plastic particles can interact with sediments through different processes like flocculation, deposition, resuspension, and burial (Wu et al., 2024b; Waldschlager and Schuttrumpf, 2019b; Shiravani et al., 2023; Andersen et al., 2021). Therefore, sediment dynamics can also affect the fate and transport of plastics, further complicating the accurate quantification of their movements. Similarly, plastic particles can interact with living organisms, leading to biofouling, where bacteria, algae, and other microorganisms attach and colonise the surface of plastic particles (Ye and Andrady, 1991; Amaral-Zettler et al., 2020). The increased size and density of the bio-fouled plastics, along with the distribution of the biofilm, can also influence their buoyancy, transport pathways, and distribution patterns (Kaiser et al., 2017; Kooi et al., 2017; Jalón-Rojas et al., 2022).

Different strategies can be employed to unravel the complexities of plastic transport and dynamics in estuaries. Field data provide valuable insights into the presence, distribution, behaviour, and evolution of plastics within estuarine environments. However, collecting comprehensive observational data is challenging, primarily due to the resource-intensive nature of monitoring efforts, sampling methods, and detection and analysis methods (Shi et al., 2023; Hale et al., 2020). For example, when conducting monitoring surveys in estuaries, it is important to consider divergent environmental factors such as river discharge, tidal patterns, and wind dynamics, as these processes can significantly affect sampling outcomes, necessitating a large number of samples (Defontaine and Jalón-Rojas, 2023). Consequently, it is also recommended to collect additional data including water level, current velocity, salinity, temperature, wind or river flow along with plastic sampling in estuaries (Defontaine and Jalón-Rojas, 2023). Remote sensing techniques, while becoming a promising solution for detecting larger plastics on the surface, are unable to capture suspended microplastics (Papageorgiou and Topouzelis, 2024). Numerical models arise as a relevant tool for understanding and predicting the transport and fate of plastic debris. Although the number of modelling studies focusing on estuary-scale plastic transport may be relatively fewer compared to those at the ocean scale (Lebreton et al., 2012; Onink et al., 2021; Wichmann et al., 2019; Tong et al., 2021), the remarkable advancements in modelling hydrodynamics and sediment

Figure 1. Time and space scales of microplastic transport processes.

dynamics within estuaries (e.g., (Stark et al., 2017; Grasso and Caillaud, 2023; Van Maren et al., 2015; Do et al., 2024)) offer a promising avenue to advance our understanding and modelling capabilities in the realm of plastic transport in these systems.

Most available studies addressing transport modelling have concentrated predominantly on microplastics, given their abundance, prevalence, mobility, and associated threats. Both Lagrangian and Eulerian approaches can be incorporated with hydrodynamic models for several purposes: (1) to identify the effect of various transport processes like advection-dispersion, settling, burial of microplastics in sediments, and their resuspension during erosive events (Jalón-Rojas et al., 2019a; Pilechi et al., 2022; He et al., 2021), (2) to assess the areas of high accumulation of microplastics like Estuarian Microplastic Maxima (EMPM)(Díez-Minguito et al., 2020), (3) to monitor the potential microplastic release from sources into the estuaries and their remobilisation, highlighting the source-sink dynamics (Sun et al., 2022), (4) to simulate the movement and dispersion of microplastics over different temporal scales, from hourly fluctuations influenced by tidal dynamics to seasonal variations, and evaluate the influence of temporally varying environmental factors and hydrodynamic characteristics (Defontaine et al., 2020; Elisei Schicchi et al., 2023), or (5) to understand the residence time and the resulting impacts of microplastics in the estuaries (Cohen et al., 2019).

Despite the significant advancements in numerical models, simulating the transport of microplastics in estuaries faces several challenges. These challenges arise from the complexities inherent in estuary dynamics and microplastic behaviour, as mentioned earlier. They are further compounded by the scarcity of observation data for validation and parameterization, as well as the high computational demands needed to capture the intricate details of estuary hydrodynamics and transport processes while accounting for multiple sources and varying particle properties.

This paper reviews modelling studies investigating the transport of microplastics in estuarine environments. The specific objectives include (1) to examine the different approaches used to simulate transport processes and dispersion trends, highlighting their respective advantages and limitations; (2) to evaluate the variety in input data, model setup configurations, model parameterizations, and calibration and validation methods used in various estuarine applications; and finally (3) to identify key challenges and provide recommendations and good practices. The final goal is to support the development of more robust methodologies and modelling strategies to advance our knowledge of microplastic dynamics at the land-ocean interface.

# 2 Modelling approaches for microplastic transport in estuaries



Process-based models used to simulate the transport of microplastics in estuaries can be classified according to different criteria: analytical vs. numerical formulations, Eulerian vs. Lagrangian frameworks, and realistic vs. idealized configurations. These classifications are not mutually exclusive, and many models combine elements from each category. In general, three main categories are used in research studies: Eulerian analytical or semi-analytical models, Eulerian numerical models, and Lagrangian numerical models. (Semi)analytical models are intrinsically idealized and use a 2DV (2D-vertical) configuration, while both Eulerian and Lagrangian numerical models can operate in both 2DH (2D-Horizontal) or 3D approaches and use realistic or idealized configurations, and may be applied to either idealized or realistic setups. In practice, realistic configurations are more commonly used in the literature. Eulerian models focus on fixed points in space and track the changes in microplastic concentrations over time at these points, whereas Lagrangian models follow individual particles or parcels of microplastics through space and time, capturing their trajectories and interactions with the flow. Additionally, a new approach for microplastic transport modelling based on the Population Balanced Equation (PBE) has been recently proposed. The PBE method describes how the number of particles with specific properties (like size) changes over time within a particle population. Figure 2 provides a schematic overview of the geometrical configurations and resulting outputs for each modelling approach.

For each approach, the implementation of microplastic transport models typically involves two steps. Initially, a hydrodynamic or hydro-sedimentary model is used to simulate key environmental variables such as water elevation, current velocity, waves, temperature, salinity, and sediment dynamics. An accurate representation of hydrodynamics is the primary prerequisite, as realistic advection and diffusion directly control the reliability of particle transport predictions. Secondly, a transport model takes the outputs from the hydrodynamic simulation to track microplastic transport. This one-way coupling between hydrodynamic and transport models can be executed either offline or online.

In this section, we explore the four modelling approaches, highlighting previous applications in estuaries. We focus on the transport processes considered in each approach, the parameterisations and formulations typically used for these processes, and the advantages and disadvantages of each approach. The upcoming sections will outline the strategies employed for model validations, input data setup, and parameter selection across different applications. Comparative discussions and recommendations will be provided at the end of the paper to highlight good practices and guide future implementations.

For this purpose, we reviewed and discussed several previous studies that have employed different modelling approaches to study microplastic transport in estuaries. Table 1 and 2 compile and compare the parameterisation and configuration of all the

modelling studies available in the literature to date. All the reviewed publications used in this paper are sourced from reputable academic databases like PubMed, ScienceDirect, Scopus and Google Scholar with a search strategy encompassing relevant keywords including "transport", "microplastics", "modelling" and "estuary". Note that the focus within this section is primarily on the transport modules, rather than the hydrodynamic components.

# 115 2.1 Eulerian analytical 2DV models

# 2.1.1 General description



Analytical models — also referred to in literature as exploratory or idealized models - can be used in 1D or 2D to simplify complex real-world scenarios, helping to understand fundamental behaviours or principles through assumptions or abstractions. These models are often computationally much less demanding than realistic ones, allowing quicker analysis to provide essential insights. In studies of microplastic transport in estuaries, analytical or semi-analytical idealized 2DV hydro-sedimentary models, commonly used for sediment transport assessments, have been adapted and utilized to investigate the role of individual estuarine hydrodynamic processes on microplastic distributions and trapping.

Eulerian analytical 2DV (EA2DV) models typically estimate longitudinal currents by resolving flow in the longitudinal (x) and vertical (z) directions in single-branch estuaries, assuming that the estuarine geometry can be parametrized by smoothed width and depth profiles (see sketch in Fig. 2). The effects of Coriolis are neglected, and density variations are assumed to be small compared to the average density, allowing for the Boussinesq approximation. The width-averaged sediment mass balance equation describes the dynamics of particle transport considering processes such as advection, turbulent diffusion, and settling while imposing a balance between the tidally averaged erosion and deposition at the bottom. As there are various analytical or semi-analytical solutions to this equation, depending on the assumptions made, this paper describes the solutions applied in previous applications of microplastic dynamics in subsection 2.1.2. EA2DV models typically calculate the subtidal concentration of particles over space (C(x,z)) or the concentration variability over both space and time, with time variability limited to a single tidal cycle (C(x,z,t)). This limitation in the time domain, along with the model's inherent simplifications, restricts the ability to account for transport processes that involve time-varying phenomena, such as flocculation, biofouling, beaching, and refloating.

Overall, EA2DV models offer an efficient means to represent the vertical distribution of microplastics, capturing stratification and layering with low computational demand. However, their negligence of lateral transport, complex geometries, and time-varying processes can limit the accuracy and applicability of EA2DV models for predicting real-world microplastic dispersion patterns.

#### 2.1.2 Applications in estuary

Studies conducted in the Ría de Vigo estuary (Díez-Minguito et al., 2020), the Guadalquivir estuary (Bermúdez et al., 2021) and the Garonne tidal River (Jalón-Rojas et al., 2024a) come under the EA2DV category (Table 1). In Ria de Vigo Estuary and Guadalquivir Estuary, stationary conditions are assumed at tidally-averaged scales and subtidal circulation was modelled

**Figure 2.** Schematic representations of an estuary, model geometries and typical outputs for Eulerian analytical (EA2DV), Eulerian numerical (ENM), Lagrangian numerical(LNM), and Population Balanced Equation (PBE) modelling approaches.

based on Talke et al. (2009), employing the classic steady and linear formulation of gravitational circulation (Hansen and Rattray Jr, 1966). Their EA2DV approach assumes a rigid lid at the surface and a no-slip condition at the bed. In these applications, the models were forced with a constant river flow at the upstream end and wind-induced shear stress at the surface, with consideration of density-driven circulation. In the Garonne tidal river, Jalón-Rojas et al. (2024a) applied the iFlow model framework (Dijkstra et al., 2017), which solves the width-averaged shallow water equations under the assumption that the amplitude of the main (semi-diurnal) tidal harmonic is small compared to the mean water depth and considering smoothed width and depth profiles. iFlow uses a perturbation technique to separate tidal and subtidal components, allowing linear equations to describe leading-order processes while incorporating non-linear effects at higher orders. iFlow model offers a wide array of options for configuring the model geometry, along with numerous selections for turbulence and salinity models (see Dijkstra et al. (2017) for further details). In the Garonne application, the model was forced with a constant river discharge at the upstream boundary and two tidal harmonics (M2, M4) at the seaward layer. As is typical for these kinds of models, the transport processes considered in three applications include advection, turbulent diffusivity, and settling.

# Parameterizations and formulations in estuarine applications



In Díez-Minguito et al. (2020) and Bermúdez et al. (2021), the spatial distribution of microplastics has been modelled from the subtidal concentration equation. In particular, the vertical distribution of particles has been derived by ensuring an equilibrium between the vertical flux of microplastics and the mixing due to turbulence following Talke et al. (2009), which is given as

$$\frac{\partial}{\partial z} [W_{Mp}C + K_v \frac{\partial C}{\partial z}] = 0 \tag{1}$$

where  $W_{Mp}$  is the settling velocity of the microplastics,  $K_v$  is the vertical diffusion coefficient, and C is the concentration of the microplastics.

Equation 1 is solved to obtain the distribution of microplastic concentration C(x,z) as a function of the average amount of microplastics at the bottom available for resuspension over the channel's length  $C_B(x)$ :

$$C(x,z) = C_B(x)^{(-W_{M_p}(z+H(x))/K_v)}$$
 (2)

where H(x) is the bottom depth with z varying between 0 at the surface and H(x) at the bottom.

In the adaptation of the iFlow model implemented in Jalón-Rojas et al. (2024a), microplastics are transported primarily as suspended load, governed by a more complete width-averaged microplastic mass balance equation including advective terms and longitudinal diffusion, expressed as:

$$\frac{\partial C}{\partial t} + u \frac{\partial C}{\partial x} + w \frac{\partial C}{\partial z} = W_{Mp} \frac{\partial C}{\partial z} + \frac{1}{B} \frac{\partial C}{\partial x} (BK_h \frac{\partial C}{\partial x}) + \frac{\partial}{\partial z} (K_v \frac{\partial C}{\partial z})$$
 (3)

where u and w are the horizontal and vertical velocities, respectively, and B is the width of the estuary. For more details about the boundary conditions, readers can refer to the original work. As previously mentioned, the variability with time is limited to single tidal cycles.

In this EA2DV approach, the erosion flux depends on the dimensionless particle availability function a(x), which describes how microplastics are distributed over the system. Both, the functions a(x) (iFlow) and  $C_B(x)$  (Talke's approach) are unknown and can be determined by ensuring a balance between the tidally averaged erosion and deposition at the bottom (the so-called morphodynamic equilibrium condition in sediment transport (Friedrichs et al., 1998; Chernetsky et al., 2010). While this equilibrium concept originates from sediment dynamics, it can be adapted to microplastic transport by treating the system as quasi-steady and focusing on the net fluxes over a tidal cycle. Although microplastics do not contribute to bed transformations, adapting this approach offers a useful framework for quantifying net accumulation zones and transport pathways under tidally averaged conditions. Readers can refer to the original works for more details.





In EA2DV transport models, it is essential to set up or calibrate three main parameters: the terminal velocity, which can either be rising or settling velocities  $W_{M}p$  and the horizontal and vertical diffusivity coefficients represented as  $K_h$  and  $K_v$ , respectively. Users can either assign their preferred values for  $W_{M}p$  or opt for state-of-the-art equations from lab experiments. In the case study of the Ría de Vigo estuary, the Stokes law was considered for assigning the value for  $W_{M}p$ , which takes into account the size and density of the particle as well as the density and viscosity of water. The simulations for Guadalquivir Estuary and Garonne tidal river utilized the advanced empirical formulations by Waldschlaeger et al. (2020). Additionally, in the case of the Garonne tidal river, the model also employed the formulations by Dioguardi et al. (2018), taking into account the particle's size, density and different shape parameters, along with water properties.

The horizontal diffusivity coefficient  $K_h$  is typically assigned as a standard constant value from literature or, less commonly, calibrated in EA2DV models. The vertical diffusivity coefficient  $(K_v)$  is commonly assumed equivalent to the vertical eddy viscosity  $A_v$ , which has been estimated using various approaches in existing applications. In the Guadalquivir estuary application,  $A_v$  is assumed as the sum of a tidally averaged component  $(A_{v0})$  and a fluctuating component  $(A'_v)$  arising from tidal straining, which accounts for the correlation between fluctuating eddy viscosity and vertical velocity shear  $\frac{\delta u'}{\delta z}$ , where u is the longitudinal current. The fluctuating part varies spatially, decreasing exponentially away from a given location. Both terms  $A_v$  and  $A'_v$  require calibration and validation.

In the Rio de Vigo estuary, the value for  $A_v$  is estimated from observational data within the region and treated as a constant value. In the Garonne tidal river,  $A_v$  is assessed in the iFlow model as a function of bottom stress and the mean depth, based on parameterizations derived from turbulence closure model  $(k - \epsilon)$  experiments.

Further details on these parameterizations in the iflow model can be found in Dijkstra et al. (2017). The described parameterizations of these applications are summarized in Table 1.

Table 1. Modelling approaches used for microplastic studies in estuaries

|                                    |                                                                          |                                                                                                                                                                                                                                                                                                                                                                                                     | Eulerian analytical 2DV models                                    |                                                                                                |                                                                                                                                                                                                                                                                                                                                                                                                                                                     |                             |
|------------------------------------|--------------------------------------------------------------------------|-----------------------------------------------------------------------------------------------------------------------------------------------------------------------------------------------------------------------------------------------------------------------------------------------------------------------------------------------------------------------------------------------------|-------------------------------------------------------------------|------------------------------------------------------------------------------------------------|-----------------------------------------------------------------------------------------------------------------------------------------------------------------------------------------------------------------------------------------------------------------------------------------------------------------------------------------------------------------------------------------------------------------------------------------------------|-----------------------------|
| Study area                         | Model                                                                    | Parameterization                                                                                                                                                                                                                                                                                                                                                                                    | Frequency/duration                                                | Sources                                                                                        | Particle properties                                                                                                                                                                                                                                                                                                                                                                                                                                 | Reference                   |
| Guadalquivir Estuary<br>(SW Spain) | tidally averaged<br>circulation model<br>based on<br>Talke et al. (2009) | • $W_r$ based on Waldschlaeger et al. (2020)<br>• $K_v = A_v$ based on previous studies and observations                                                                                                                                                                                                                                                                                            | one tidal cycle                                                   | -                                                                                              | PE films (a) $W_r = 0.0046$ m/s (b) $W_r$ varying from 0.001 to 0.010 m/s                                                                                                                                                                                                                                                                                                                                                                           | Bermúdez et al. (2021)      |
| Ria de Vigo Estuary<br>(NW Spain)  | tidally averaged<br>circulation model<br>based on<br>Talke et al. (2009) | • $W_{Mp}$ based on Ballent et al. (2013),<br>Khatmullina and Isachenko (2017) and Wegner et al. (2012)<br>• $K_v = A_v$ based on observations                                                                                                                                                                                                                                                      | one tidal cycle                                                   | dimensionless dimensionless microplastics availability function* which describes microplastics | (a)Pellets $W_{Mp} = \pm 0.02 \text{ m/s}$<br>(b)Fibers $W_{Mp} = \pm 0.0067 \text{ m/s}$<br>(c)Fragments $W_{Mp} = \pm 1.31 \times 10^{-7} \text{ m/s}$                                                                                                                                                                                                                                                                                            | Díez-Minguito et al. (2020) |
| Garonne tidal river<br>(SW France) | iflow model                                                              | • $W_{MF}$ based on Waldschlaeger et al. (2020), Dioguardi et al. (2018); • $K_v$ based on simplified parametrization based on 1D k- $\epsilon$ model                                                                                                                                                                                                                                               | Two single<br>tidal cycles                                        |                                                                                                | (a)particles with $W_s = 0$ m/s (b)particles with $W_{Mp} = 0.0005$ m/s (c)particles with $W_{Mp} = 0.002$ m/s                                                                                                                                                                                                                                                                                                                                      | Jalón-Rojas et al. (2024a)  |
|                                    |                                                                          |                                                                                                                                                                                                                                                                                                                                                                                                     | Eulerian numerical models                                         |                                                                                                |                                                                                                                                                                                                                                                                                                                                                                                                                                                     |                             |
| Study area                         | Model                                                                    | Parameterization                                                                                                                                                                                                                                                                                                                                                                                    | Frequency and duration                                            | Sources                                                                                        | Particle properties                                                                                                                                                                                                                                                                                                                                                                                                                                 | Reference                   |
| Adour Estuary<br>(SW France )      | Telemac 3D                                                               | • $W_{MP}$ based on<br>Kowalski et al. (2016),<br>Khatmullina and Isachenko (2017);<br>• $K_v = A_v$ based on<br>Prandtls ML model                                                                                                                                                                                                                                                                  | a single release with<br>10 g/l concentration;<br>duration=9 days | Maincity                                                                                       | (a)PS(0.5 $mm$ )<br>$W_{Mp}$ =0.004 $m/s$<br>(b)PCL(4.9 $mm$ )<br>$W_{Mp}$ =0.127 $m/s$<br>(c)neurally-buoyant<br>paricles(3 mm)                                                                                                                                                                                                                                                                                                                    | Defontaine et al. (2020)    |
| Weser Estuary (S North sea)        | Delft 3D Flow -<br>WAQ FSK MP<br>tracking model                          | • $W_{MP}$ based on Kooi et al. (2017), Dietrich (1982); • $K_v = A_v$ , based on $k - 1$ model; • Flocculation based on emphirical formulations as a function of sediment concentration; • Deposition based on Partheniades (1965), Krone (1962); Krone (1962); • Resusupension based on modified Wu et al. (2018); • Biofouling based on Kooi et al. (2017) and from simplified ecological model. | (not specified)                                                   | WWTP,<br>atmospheric<br>wastes                                                                 | (a)PE(buoyant)<br>(b)PP(buoyant),<br>(c)PS<br>( $W_s = 2.7 \times 10^{-6} \text{ m/s}$ )<br>(d)PVC<br>( $W_s = 2.38 \times 10^{-5} \text{ m/s}$ )<br>(c)PUR<br>( $W_s = 1.17 \times 10^{-5} \text{ m/s}$ ),<br>(f)PET<br>( $W_s = 2.04 \times 10^{-5} \text{ m/s}$ ),<br>(g)PC<br>( $W_s = 9.49 \times 10^{-5} \text{ m/s}$ ),<br>(g)PC<br>( $W_s = 9.49 \times 10^{-5} \text{ m/s}$ ),<br>(h)PMIMA<br>( $W_s = 1.04 \times 10^{-5} \text{ m/s}$ ), | Shiravani et al. (2023)     |

PC:Polycarbonate; PMMA:Polymethylmethacrylate; \* unknown and can be determined by imposing the morphologic equilibrium condition which is assumed that the amount of microplastic concentration in Wr.: Rising velocity; WWTP: Waste Waste Water Treatment Plant; PE:Polyethylene; PS:Polystyrene; PP:Polypropylene; PO:Polyvinyl chloride; PUR:Polyurethane; PET:Polyethylene terephthalate; the system is a constant and look for equilibrium distribution of microplastics in the estuary. Equilibrium here means a balance between the tidally averaged erosion and deposition at the bottom.

Table 2. (Continued)

|                                                                  |                                                      |                                                                                                                                                                                                                                       | Lagrangian numerical models                                                                                                                                    |                                                 |                                                                                                                          |                       |
|------------------------------------------------------------------|------------------------------------------------------|---------------------------------------------------------------------------------------------------------------------------------------------------------------------------------------------------------------------------------------|----------------------------------------------------------------------------------------------------------------------------------------------------------------|-------------------------------------------------|--------------------------------------------------------------------------------------------------------------------------|-----------------------|
| Study area                                                       | Model                                                | Parameterization                                                                                                                                                                                                                      | Frequency and duration                                                                                                                                         | Sources                                         | Particle properties                                                                                                      | Reference             |
| Changjiang River Estuary<br>(E.China)                            | SCHISM                                               | • $W_{MP}$ based on Kowalski et al. (2016) Waldschlager and Schutrumpf (2019a) Khatmullina and Isachenko (2017)                                                                                                                       | Frequency=6hours<br>Duration not specified                                                                                                                     | Industrial<br>locations                         | (a)PP $(W_s=0.01\text{m/s})$ , (b)PE $(W_s=0.0065\text{m/s})$ , (c)PS $(W_s=0.0065\text{m/s})$ , $(W_s=0.015\text{m/s})$ | Sun et al. (2022)     |
| Ria de Vigo Estuary<br>(NW Spain)                                | Delft-3D flow<br>Delft-PART                          | • $W_{M,p}$ based on<br>Stokes law<br>• $K_v = A_v = \text{constant}$<br>• beaching based on MCA<br>no refloating                                                                                                                     | (a)continous release<br>for 3 days<br>duration=35days<br>(b)instantaneous release<br>during 4 tidal cycle                                                      | different<br>WWTPs                              | Particles of sizes $10\mu \text{m}$ , $1\text{mm}$ ,2.5mm, $5\text{mm}$                                                  | Sousa et al. (2021)   |
| Lower Saint John<br>river Estuary<br>(E.Canada)                  | CaMPSim-3D -<br>Telemac 3D                           | • modified RWM • $W_{MP}$ based on Dellino et al. (2005) • $K_h$ constant $K_v$ (a) constant $K_v$ (a) constant on hydrodynamic model                                                                                                 | (a)instantaneous release with 0.1 sec frequency and duration=100sec (b)throughout release with 15 minute frequency with (1)duration=15 days (2)duration=7 days | Effluent<br>plume<br>within<br>the estuary      | Neutrally<br>buoyant<br>particles                                                                                        | Pilechi et al. (2022) |
| Santos Estuary<br>(S.E.Brazil)                                   | Lagrangian PTM<br>(Harari and Gordon;<br>2001) - POM | • no sinking • $K_{h_i}$ using RWM based on Rubinstein and Kroese (2016) • beaching without considering refloating                                                                                                                    | random release<br>for one year                                                                                                                                 | river mouth,<br>beach and<br>offshore<br>zones) | Pellets                                                                                                                  | Gorman et al. (2020)  |
| Yangtze Estuary<br>(E.China)                                     | MIKE21 -<br>Random walk<br>Lagrangian tracking model | • $W_{MP}$ based on formulae developed of for $C_D$ for $C_D$ for $C_D$ has based on hydrodynamic model • flocculation based on conditional settling velocities • biofouling with constant rate and based on Chubarenko et al. (2016) | duration=3 periods of<br>1-month duration<br>(frequency not specified)                                                                                         | river                                           | Fibrous PE,<br>PP, PVC                                                                                                   | Shen et al. (2022)    |
| Río de la Plata Estuary,<br>(Border of Argentina<br>and Uruguay) | MARS -<br>TrackMPD                                   | • $W_{MP}$ for sphere based onZhiyao et al. (2008) • $W_{MP}$ for cylinder based on $W_{MP}$ for cylinder based on $K_{V}$ as constant • Beaching based on MCA no refloating                                                          | duration=1 month<br>frequency=1 hour                                                                                                                           | river mouth<br>WWTP                             | $PP(W_M p=0)$ , PS spheres, PS cylinders                                                                                 | Shen et al. (2022)    |
| Bach Dang estuary<br>(Vietnam)                                   | Delft 3D- PART model                                 | • $W_M p$ based on empirical formulations (not specified) • $K_h = A_h = \text{constant}$ from hydrodynamic model • $K_v = A_v = \text{constant}$ from hydrodynamic model                                                             | continuous release<br>(duration not specified)                                                                                                                 | river mouth                                     | PE(4mm)                                                                                                                  | Le et al. (2022)      |
|                                                                  |                                                      |                                                                                                                                                                                                                                       |                                                                                                                                                                |                                                 |                                                                                                                          |                       |

MCA: Monte Carlo Approach

#### 2.2 Eulerian numerical models

#### 2.2.1 General description





Eulerian hydrodynamic and transport models are well-suited for simulating large-scale simulations, realistically capturing complex flow dynamics and transport trends in the study of microplastic dispersion. Eulerian numerical models (ENM)predict how microplastic concentrations change over time and space within the flow field. This approach is particularly well-suited for suspended particles in scenarios where all microplastics exhibit uniform behaviour, either passive or similar to sedimentary particles, although discrete classes of behaviours may also be envisaged in some of these models.

In ENMs, fluid flow is generated within an Eulerian framework where the fluid properties, like flow velocities, temperature, and salinity, are defined at fixed points in space. The fluid motion is described by the Navier-Stokes equations, which express the conservation of mass and momentum, according to various levels of simplification. The equations are numerically solved to obtain the fluid's velocity field, providing a detailed representation of flow dynamics in the simulated domain. Hydrodynamic forcing parameters typically include tide levels or tidal constituents at the model's sea boundaries, river discharge data at river boundaries, and initial salinities. Additional forcings such as initial/boundary currents, temperature, waves and wind may also be required depending on the applications and the study site.

ENMs can be employed with several discretisation methods and grid structures based on the domain and the application (Winterwerp et al., 2022). The importance of using sigma layers for vertical grids in sediment transport studies is well documented by Cancino and Neves (1999) and Winterwerp et al. (2022), as they allow for an accurate representation of sediment dynamics in water columns. This approach is particularly relevant for microplastic transport studies, where sediment transport models can be adapted to study the dynamics and behaviours of microplastic particles.

Microplastic concentrations in ENMs are calculated spatially and temporally by solving the advection-diffusion equation, akin to suspended sediment transport models. This equation describes the interplay between advection and diffusion on particle movement and dispersal within the system, with vertical advection accounting for the terminal velocities (settling or rising) of the microplastic particles. In its conservative form, the advection-diffusion equation used in ENMs is,

$$\frac{\partial C}{\partial t} + u(\frac{\partial C}{\partial x}) + v(\frac{\partial C}{\partial y}) + (w + W_{Mp})(\frac{\partial C}{\partial z}) - \frac{\partial}{\partial x}(K_h(\frac{\partial C}{\partial x})) - \frac{\partial}{\partial y}(K_h(\frac{\partial C}{\partial y})) - \frac{\partial}{\partial z}(K_v(\frac{\partial C}{\partial z})) = 0 \tag{4}$$

where  $\frac{\partial C}{\partial t}$  is the change in particle concentration over time and  $(\frac{\partial C}{\partial x}), (\frac{\partial C}{\partial y})$  and  $(\frac{\partial C}{\partial z})$  are the spatial change in concentrations with velocity field u, v and w in x, y and z directions, respectively. An extra terminal velocity term  $W_{Mp}$  is included in the equation to calculate the vertical transport.

Similar to analytical models, the implementation of ENMs requires choices to be made in the parametrization of the  $W_M p$ , 230  $K_h$  and  $K_v$ . The horizontal and vertical turbulent mixing of particles is represented through  $K_h$  and  $K_v$ , respectively, which can be a constant value over time and space or spatially and temporally varying values. ENMs usually incorporate various types of turbulence closure models  $(k - \epsilon, k - \omega, k - l)$ , Smagorinsky model, Prandtl's mixing length model and so forth) to calculate vertical eddy viscosities  $(A_v)$  and represent mixing processes.

Additionally, the representation of erosion processes of settling particles or flocculation can be easily taken into account by following the available parametrizations for suspended sediment transport. In the case of flocculation, further experimental research is required to refine these parameterizations and enhance their applicability to microplastic dynamics. ENMs typically incorporate erosion and deposition processes using the classical approaches proposed by Partheniades (1965), Krone (1962) and Nvm and Owen (1972), which were initially designed for cohesive sediment particles. The deposition and erosion fluxes of microplastics are considered as controlled by the local bed shear stress relative to their respective critical threshold, with deposition occurring when bed shear stress is lower than the critical deposition threshold and erosion occurring when the bed shear stress exceeds a critical erosion threshold. The erosion-deposition equations are normally used when there is an interaction of settling microplastic particles with the sediment bed. A challenge of the ENM approach is to calibrate or set up multiple parameters, such as critical shear stress for deposition and erosion, deposition flux or erosion flux, when observations on the erosion processes of microplastics are still scarce.

ENMs are primarily employed to analyse the particle dynamics in the water column, typically applied to neutrally buoyant or non-buoyant particles. Fine transport processes with coastlines, such as beaching, have generally not been accounted for up to now, while potentially implementable with fine spatial resolution and accurate representation of the wetting and drying dynamics. Several factors discussed in Section 2.5 - such as the challenges in representing floating particle dynamics within Eulerian frameworks, their inability to explicitly resolve source-to-sink pathways, and the widespread adoption of Lagrangian approaches in global-scale plastic transport studies - may have limited the utilisation of ENMs in the study of microplastics.

# 2.2.2 Applications in estuary








In this section, we describe and analyse the methodologies employed in two available studies: Shiravani et al. (2023) and Defontaine et al. (2020), which used Delft-3D and Telemac-3D Eulerian models, respectively, to explore microplastic dynamics in estuaries (Table 1). Shiravani et al. (2023) investigated the aggregation of microplastic particles with fine sediments and microorganisms in the Weser estuary. The microplastic settling formulations are refined based on suspended fine sediment concentrations, enabling an exploration of the interaction with sediments and an assessment of the microplastic entrapment within the water column and sediments, particularly in the Estuarine Turbidity Maximum (ETM). Defontaine et al. (2020) investigated the distribution of suspended microplastics in the Adour estuary, applying the ENM to simulate the vertical and longitudinal movement of suspended particles in response to tidal currents.

# Parametrizations and formulations in estuarine applications

The two analyzed applications considered advection, turbulent mixing, setting, deposition and resuspension. The parameterization of each process, along with the choice of key parameters such as  $W_M p$ ,  $K_h$ ,  $K_v$ , deposition and erosion fluxes, and critical shear stress for deposition and resuspension, are crucial for interpreting and discussing the resulting transport trends. Additionally, Shiravani et al. (2023) included the potential flocculation of microplastics with fine sediments and biofouling. All these aspects are described in this section and summarized in Table 1.

For the parametrization of  $W_M p$ , Defontaine et al. (2020) provided direct values based on available literature like Kowalski et al. (2016) and Khatmullina and Isachenko (2017). In Shiravani et al. (2023),  $W_M p$  was computed using the formulation proposed by Kooi et al. (2017) according to (Dietrich, 1982), a formulation developed for natural particles.

Shiravani et al. (2023) assessed the microplastic-fine sediments interaction and included the flocculation process by coupling the hydro-sedimentary model with the water quality model WAQ. In this model, the particles's  $W_{M}p$  varied as a function of fine sediment concentrations. The thresholds in sediment concentration were estimated by considering some experimental observations provided by Andersen et al. (2021) for PVC particles and Oberrecht (2021). To examine the impact of biofouling, the growth rate of microalgae is incorporated in the calculation of microplastic settling velocities and density following the approach by Kooi et al. (2017). However, instead of using the empirical equations from the original study, simulated values based on a simplified ecological model were used to estimate the diatom concentrations in the estuary. More explanations on these parameters can be found in Shiravani et al. (2023).




Regarding particle diffusivity coefficient, a constant  $K_h$  is assigned by Defontaine et al. (2020) while no information on this parameter is reported in Shiravani et al. (2023).  $K_v$  is assumed to be the same as  $A_v$  in both studies. However, each application used a different approach for representing  $A_v$ . In the case of the Adour estuary, Prandtl's mixing length theory was applied (Defontaine et al., 2020), where  $A_v$  is expressed as a function of the vertical velocity gradient and the mixing length. Furthermore, the authors incorporated a damping function to account for the damping of turbulent mixing caused by density stratification. The Weser estuary study (Shiravani et al., 2023) applied the k-l model, which uses the transport equations for turbulent kinetic energy (k) and the length of the turbulent eddy (l) to calculate  $A_v$ .

Concerning the interactions with the bottom, Defontaine et al. (2020) do not provide details on the parameterization of deposition and resuspension fluxes in their study. The microplastic deposition flux in the study of Shiravani et al. (2023) is given by the classical equations. In this application, critical shear stress for deposition is assumed to be the same as that of nearby fine sediments, with a constant value based on previous studies. For the resuspension flux, the authors adopted the equation from Wu et al. (2018), redefining the critical bed shear stress for mixed sediments as the critical resuspension shear stress for microplastics in mixed sediments ( $\tau_{c_r,Mp,m}$ ) as

$$\tau_{c_r,Mp,m} = \tau_{c_r,Mp,s} + (\tau_{c_r,Mp,pm} - \tau_{c_r,Mp,s}) \exp[-f_1(\frac{p_s}{p_m})^{f_2}]$$
 (5)

where  $\tau_{c_r,Mp,s}$  is the critical resuspension shear stress for microplastics in sand (Waldschlager and Schuttrumpf, 2019a),  $\tau_{c_r,Mp,pm}$  is the critical resuspension shear stress for microplastics in pure mud,  $p_s$  and  $p_m$  are the percentage of sand and fine sediment/mud in mixed sediment respectively and  $f_1$  and  $f_2$  are calibration parameters which are estimated by comparing the model results with the available observation data.

In this equation, to address the unknown term  $\tau_{c_r,Mp,pm}$ , the authors proposed it as a function of mud porosity,  $\rho_{Mp}$ , mud density, and the microplastic size.

Then the resuspension flux for microplastics is defined using the  $\tau_{c_r,Mp,m}$  in a similar way as deposition and erosion flux by using a threshold value for  $\tau_{c_r,Mp,m}$  and the bed shear stress.

# 2.3 Lagrangian Numerical models

# 2.3.1 General description



Unlike ENMs, Lagrangian particle tracking models (LNM) monitor the position of individual particles over time (Van Sebille et al., 2018). These models trace the trajectories of particles as they move through the fluid and can incorporate specific particle behaviours and various transport processes (Jalón-Rojas et al., 2019a). LNMs can leverage inputs from the ENMs or observations, such as current velocities, through either online or offline coupling. Online coupling integrates hydrodynamic ENM and LNMs simultaneously, enabling time-resolved flow action on the particle field during simulation. By contrast, offline coupling runs ENMs and LNMs separately, with ENMs providing precomputed flow fields to the LNM after the completion of its simulation. Both coupling methods are powerful approaches to gain valuable insights into the dynamic behaviour and dispersion patterns of microplastics in dynamic flow systems. According to our literature review, the vast majority of previous modelling studies on microplastic dispersion in estuaries have used the LNM approach as observed in Table 2.

The particle movement in LNM approaches is typically divided into a deterministic or resolved component and a turbulent, unresolved contribution. The resolved component represents the advection and is calculated by integrating the (spatially-and temporally-varying) velocity field. Unresolved contributions are typically addressed through stochastic terms, which are commonly represented by the diffusive component, accounting for unresolved scales of the flow (Jalón-Rojas et al., 2019b; Pilechi et al., 2022; Shen et al., 2022). Additionally, settling/rising velocities can be incorporated to account for gravitational and buoyant forces acting on particles, influencing their vertical movement within the fluid environment (Jalón-Rojas et al., 2019b). The basic equations representing the particle movement by incorporating the advection, diffusion and settling/rising transport terms in LNMs are given as,

$$dX(t) = dX_{adv}(t) + dX_{dif}(t) = U(x, y, z, t)\Delta t + dX'(t)$$

$$\tag{6}$$

$$dY(t) = dY_{adv}(t) + dY_{dif}(t) = V(x, y, z, t)\Delta t + dY'(t)$$
(7)

$$dZ(t) = dZ_{adv}(t) + dZ_{dif}(t) + dZ_{tv}(t) = W(x, y, z, t)\Delta t + dZ'(t) + W_{Mp}\Delta t$$
 (8)

where the advective terms  $dX_{adv}(t)$ ,  $dY_{adv}(t)$  and  $dZ_{adv}(t)$  represent the particle movement by the velocity fields U, V and W in the x, y and z directions, respectively, over time t, typically provided by hydrodynamic ENMs. The diffusion terms  $dX_{dif}(t)$ ,  $dY_{dif}(t)$  and  $dZ_{dif}(t)$  represent the motion of the particles due to turbulent diffusion in the corresponding directions x, y and z and are represented in the equation using the random components dX'(t), dY'(t) and dZ'(t). The sinking/rising displacement of microplastics,  $dZ_{tv}(t)$ , is calculated as  $W_{Mp}\Delta t$ . dX(t), dY(t) and dZ(t) are the overall particle movement due to both advection and diffusion in x, y and z directions, respectively.

The advection term in LNMs can also incorporate the direct impact of wind or windage on particle transport. This effect can be particularly relevant for floating macroplastics with a high surface exposed to wind (Critchell and Lambrechts, 2016; Jalón-Rojas et al., 2019a; Stagnitti and Musumeci, 2024).

In estuarine studies, the diffusion stochastic term mainly represents turbulent mixing and has been typically calculated using random walk models. The classic random walk model equation used to represent the diffusion term is given as

$$d_{dif}(t) = R\sqrt{2K\Delta t} \tag{9}$$

where R represents the random change of the particle position derived from a random number between -1 and 1, and K is the diffusivity coefficient, which can either be horizontal  $(K_n)$  or vertical  $(K_n)$ .

As with previous approaches, the choice of  $W_{Mp}$  depends on the specific characteristics of the microplastic particles and the study's objectives. Determining particle velocities in LNMS is typically flexible, allowing user-defined values or empirical formulations obtained through laboratory experiments. Empirical formulations can be implemented "off-line". However, some models, such as TrackMPD (Jalón-Rojas et al., 2019a)) and CaMPSim-3D ((Pilechi et al., 2022), already incorporate state-of-the-art formulations to directly calculate velocities as a function of the particle physical parameters or other processes such as biofouling, flocculation or degradation. Additionally, LNMs can effectively include other transport processes such as beaching, refloating, and fragmentation.

# 2.3.2 Applications in estuary






Our literature review compiles seven studies exploring microplastic transport and dynamics in estuaries using various LNMs (Table 2). Of these, four employed offline coupling methods (Pilechi et al., 2022; Gorman et al., 2020; Shen et al., 2022; Elisei Schicchi et al., 2023) while three utilized online coupling (Sousa et al., 2021; Sun et al., 2022; Le et al., 2022). The LNMs used included CaMPSim, TrackMPD, Delft-PART, SCHISM and the model by Harari and Gordon (2001). These studies primarily aimed at investigating seasonal trends in microplastic trajectories, the influence of environmental forcings such as tides, winds and river discharge on transport trends, and the effect of beaching or flocculation with sediments on microplastic dynamics. A study employed backward trajectories for identifying the potential sources of microplastics in the East China Sea, including estuaries (Sun et al., 2022).

#### Parametrizations and formulations in estuarine applications

LNMs can apply various advection and diffusion schemes. One of the analyzed studies, Pilechi et al. (2022), did a comparative analysis of advection schemes to resolve the advection term in Equations 6 and 7 using a case study of microplastic transport in the Saint John River Estuary. The tested schemes include Euler, second, third, and fourth-order Runge-Kutta (RK) schemes, second and third-order Total Variation Diminishing (TVD) Runge-Kutta schemes, and the Second-order Adams-Bashforth scheme. Among these, the RK4 method emerged as the most accurate advection scheme when validated with the analytical test cases with passive particles, followed by TVD3 and RK2, respectively, with the Euler scheme as the least precise method. While the advection schemes utilized for tracking microplastic were unreported in most of the studies analysed here (Table 2), some LNMs such as CaMPSim (Pilechi et al., 2022) and TrackMPD (Jalón-Rojas et al., 2019a; Elisei Schicchi et al., 2023) are flexible in selecting the preferred method, with RK4 being the preferred choice for microplastic transport studies at regional and oceanic scales (Wu et al., 2024a; Lobelle et al., 2021; Jalón-Rojas et al., 2019a).







As discussed above, the studies by Sun et al. (2022) and Elisei Schicchi et al. (2023) in estuaries of the East China Sea and Río de la Plata Estuary respectively included the effect of windage by incorporating wind velocity into the advection part of the equations 6 to 8. In the latter study, the LNM TrackMPD has been utilized which can include the direct effect of wind using  $(U_w, V_w, W_w)$  and  $(U_c, V_c, W_c)$  instead of U, V, W in 6 to 8 where  $(U_w, V_w, W_w)$  are wind velocities and  $(U_c, V_c, W_c)$  are current velocities (Jalón-Rojas et al., 2019a).

The majority of the estuarine studies using LNMs (Table 2) considered the typical random walk model (equation 10) representing the diffusion component. However, alternative random walk models can also be applied; for instance, Pilechi et al. (2022) (Saint John River estuary) used a modified random walk model that calculates diffusivity using a spatiotemporal varying diffusion coefficient with the following formulation

$$d_{dif}(t) = \frac{\partial K}{\partial X} + \frac{R}{\Delta t} \left\{ \sqrt{2K[X(t) + \frac{1}{2}\frac{\partial K}{\partial X}\Delta t]\Delta t} \right\}$$
(10)

where X is the particle's position and K is the diffusivity coefficient at X at time t. This formulation can be particularly appropriate in estuarine studies as these systems can be subject to strong spatio-temporal variability of vertical mixing (Simpson et al., 1990; Burchard, 2009), potentially influencing microplastic transport.

LNMs can be flexible in defining the diffusivity coefficients. Pilechi et al. (2022) confirmed the ability of their LNM to model diffusivity by performing diffusion-only simulations using the classic random walk approach and compared the results with analytical solutions. Most of the estuarine applications are considered to have a constant  $K_h$ . Depending on the study objectives and the specific LNM,  $K_v$  was either a user-defined constant value (Sousa et al., 2021; Le et al., 2022) or a constant or time-space-varying value derived from the associated hydrodynamic model (Shen et al., 2022; Pilechi et al., 2022; Sun et al., 2022). The studies using this last approach assumed that eddy diffusivity and eddy viscosity are equivalent, similar to ENMs applications, assuming that momentum and particles diffuse at the same rate.

Similarly to ENMs, the microplastic terminal velocity is a key parameter to be defined. Previous applications of LNMs used a variety of state-of-the-art empirical equations based on laboratory experiments. For instance, the study at Rio de La Plata

Estuary (Elisei Schicchi et al., 2023) used the formulations by Zhiyao et al. (2008) for spherical particles and by Khatmullina and Isachenko (2017) for cylindrical particles, provided in the TrackMPD model to calculate settling velocities as a function of particle density and size. TrackMPD also incorporates the empirical formulations from Waldschlager and Schuttrumpf (2019a), developed for pellets, fragments, and fibres longer than 0.3 mm, and Dellino et al. (2005), which showed good performance for sheet-shaped microplastics higher than 1 mm (Jalón-Rojas et al., 2022).








In the study of the Yangtze estuary, Shen et al. (2022) used the formulation developed by Zhu et al. (2017) to calculate the settling velocity for non-spherical particles. In this formulation, the drag coefficient depends on a particle shape factor (ASF) and the particle Reynolds number  $R_e$ . While Sousa et al. (2021) (Ria de Vigo estuary) employed the standard Stokes' law, Sun et al. (2022) (estuaries in the East China Sea) used three different empirical formulations, (Waldschlager and Schuttrumpf, 2019a; Khatmullina and Isachenko, 2017) and Kowalski et al. (2016) for Polyethylene (PE), Polystyrene (PS) and Polypropylene (PP) microplastics respectively. Pilechi et al. (2022) (Saint John River estuary) used a constant value in its study with the CaMPSim-3D LNM. This model also includes the empirical formulation by Dellino et al. (2005) to calculate both settling and rising velocity with drag coefficient by Dioguardi et al. (2018). However, in the given study, these formulations are not activated as they only considered the neutrally buoyant particles.

Regarding other transport mechanisms such as flocculation and biofouling, only the study by Shen et al. (2022) incorporated these processes. In this study, the microplastic settling velocity depends on a flocculation factor that varies with sediment concentration, where  $W_{Mp}$  is modified with the flocculation factor, which increases with sediment concentration, enhancing the settling in moderate turbid waters and reaching a maximum limit when sediment becomes too dense. However, the formulation used for the study remains uncertain, as the parameterisation lacks robust empirical support and requires more experimental studies to accurately capture the dependencies of sediment-microplastic interactions.

To include biofouling, Shen et al. (2022) used the approach proposed by Jalón-Rojas et al. (2019a) in TrackMPD, which employs Chubarenko et al. (2016)'s formulation for cylindrical microplastics and assumes an increasing biofilm thickness over time as a function of a biofilm rate. This study used the same values proposed in Jalón-Rojas et al. (2019a) for the biofilm-related parameters. TrackMPD also incorporates the temporal evolution of terminal velocities, accounting for biofilm effects such as changes in density or size increase rates (Jalón-Rojas et al., 2024b). However, more experimental studies are needed to improve biofouling parametrisations. LNMs such as CaMPSim and TrackMPD have also included some exploratory formulations to take into account particle degradation as a function of a degradation rate (Jalón-Rojas et al., 2019a; Pilechi et al., 2022; Bigdeli et al., 2022).

Beaching-refloating is another significant transport mechanism that can play a crucial role in the transport and fate of floating microplastics. Among the analysed studies (Table 2), only two considered beaching, Elisei Schicchi et al. (2023) (TrackMPD, Rio de la Plata Estuary) and Gorman et al. (2020) (an LNM developed by Harari and Gordon (2001), Santos estuary). Both studies considered particles as permanently stranded upon reaching the land and neglected the refloating process.

Sousa et al. (2021) (Ria de Vigo estuary) and other previous LNM studies at coastal (Jalón-Rojas et al., 2019a) and regional (Liubartseva et al., 2018) scales, used a simplified method to account for refloating using the Monte Carlo approach. This method considers that the probability of refloating decreases exponentially with time due to interaction with the coastline.

Jalón-Rojas et al. (2019a) proposed to evaluate this condition uniquely at high tide, a consideration that may be particularly relevant for estuarine studies. However, further fundamental research is needed to improve refloating parameterisations.

# 2.4 A new approach based on the Population Balance Equation




A new modelling approach has recently been proposed to study microplastic transport in estuaries (Shettigar et al., 2024). The method is based on the Population Balance Equation (PBE), which incorporates a deposition sink term alongside advection-diffusion terms. PBE is widely used across various disciplines such as modelling the oxygen content of bubbles, wastewater treatment to simulate flocs sizes, and atmospheric aerosol distribution. Its main advantage lies in its ability to capture the dynamics of microplastics across a continuum of particle sizes (or other particle parameters). While PBE approach is conceptually similar to Eulerian approaches by estimating the time and space evolution of tracer concentration through an advection-diffusion equation, it employs a continuous distribution of particle size, in contrast to ENM or LNM, which use discrete particle classes.

The PBE method employs the Number Density Function (NDF) to represent the continuous changes in particle properties within a population due to advection and diffusion, and can also incorporate discontinuous changes from aggregation and breakage. In this framework, particles are characterised by internal properties (e.g., size, volume, mass) and external (spatial and temporal) coordinates. The transport equation for the PBE method in terms of NDF, including source and sink terms, is given as,

$$\frac{\partial}{\partial t}(hn\xi) + \frac{\partial}{\partial x}(hUn(\xi)) + \frac{\partial}{\partial y}(hVn(\xi)) - \frac{\partial}{\partial x}(hK\frac{\partial}{\partial x}n(\xi)) - \frac{\partial}{\partial y}(hK\frac{\partial}{\partial y}n(\xi))$$

$$= Source(\xi; x, y, t) - Sink(\xi; x, y, t)$$

where  $n(\xi)$  is the NDF of the particle population, with the internal coordinate  $\xi$  representing the particle size in the spatial (x,y) and temporal coordinates t. U and V are the flow velocities, h is the water depth and K is the eddy diffusivity.

The source-sink terms in the equation are assumed to represent deposition. In the study by Shettigar et al. (2024), only the sink/deposition term for microplastics is considered, excluding the erosion effect. Therefore, in equation (11), the source term is assumed to be zero, while the sink term is modelled as a function of  $W_{M_p}$ , bed shear stress and shear stress for deposition, similar to the classical approaches decsribed for deposition in Section 2.2.1. In this study,  $W_{M_p}$  is calculated as a function of particle size using the formulation from Turton and Clark (1987). However, the authors did not provide details about the approach or values to parametrise K and shear stress for deposition. Shettigar et al. (2024) compared the PBE method with the ENM using discrete classes of microplastic sizes, revealing that PBE requires less computational time. PBE model is particularly effective in systems where particle characteristics vary continuously, as it avoids the limitations of ENMs, which

typically rely on fixed particle classes and may struggle to capture shifts in particle distributions due to spatial or temporal variations.

The PBE method is still in its early stages, but it shows promise due to its ability to account for all particle sizes (or other particle parameters). However, it faces challenges similar to those of ENMs, including the establishment of initial conditions for concentration C(t=0) at multiple sources, or the parameterization of erosion-deposition fluxes. The PBE method must still demonstrate efficiency when modelling longer time scales across large-scale realistic domains.

# 2.5 Advantages and limitations of different modelling approaches






The choice between the different microplastic transport model approaches depends on the specific research questions to be addressed and the computational resources. The availability of pre-existing hydrodynamic models for the study site can also influence this choice, as leveraging existing implementations may offer advantages in terms of data availability, calibration, and validation. The existence of an Eulerian or Lagrangian transport module associated with the hydrodynamic model, as seen in modelling frameworks such as Telemac, Delft 3D, SCHISM, and CamPSIm- 3D, can significantly influence the decision-making process. However, it is essential to acknowledge that each method comes with its advantages, strengths, weaknesses, and limitations. In this section, we summarize these points for the three main methods. Regarding the new approach based on PBE, as discussed in 2.4, it is still at an early development stage, and for that reason, we did not delve further into it here. However, it seems promising for simulating the transport of a wide spectrum of particles while facing challenges similar to those of ENMs.

**Eulerian idealized models:** Idealized studies offer valuable insights by simplifying complex systems into more manageable representations, allowing for a deeper understanding of fundamental processes and mechanisms. For this kind of study, EA2DV models reduce the computational cost while still capturing the essential transport processes. This enables extensive simulation plans across a range of scenarios and allows for numerous sensitivity tests, which can be specifically relevant in microplastic modelling due to the uncertainty associated with some model parameters and the diversity of particles. A key strength of the EA2DV approach is the possibility of isolating and evaluating the relative influence of each transport process on microplastic transport and trapping trends. For instance, Bermúdez et al. (2021) successfully assessed the significance of river discharge, wind, density-driven circulation and tidal straining on the transport and trapping of microplastics at different estuarine regions.

Even though EA2DV can be effective for getting a primary understanding of transport processes and particle dynamics, it comes with significant limitations, the most notable being its oversimplification, from the system geometry to the environmental forcing. EA2DV usually fail to capture small-scale processes and complex changing 3D patterns of current flow over time. It should be noted that most EA2DV simulate subtidal conditions (Bermúdez et al., 2021; Díez-Minguito et al., 2020) or time-varying conditions over a single tidal cycle (Jalón-Rojas et al., 2021) for a given scenario of environmental forcings. The EA2DV approach is therefore not well-suited for long-term simulations that capture changing environmental forcings. The idealised models typically do not represent various transport processes, including lateral transport, beaching, flocculation, biofouling and degradation. In conclusion, EA2DVs are best suited for providing initial insight into microplastic transport mechanisms, exploring scenarios or testing hypotheses, rather than for detailed, accurate predictions.

Eulerian and Lagrangian realistic models: ENMs and LNMs can be classified as high-complexity models that incorporate realistic representations of estuarine dynamics and transport processes, offering detailed insights into the behaviour and transport patterns of microplastics within complex environmental systems. Both types of models provide useful macroscopic insights into microplastic trends, such as the changes in the microplastic concentration or position over time within large-scale systems influenced by numerous environmental forcings. Even if it depends on the specific model, both approaches can allow the incorporation of discrete classes of particles to represent a broad variety of microplastic properties.







LNMs offer several strengths, such as their ability to track individual particle trajectories, providing a high-resolution understanding of microplastic pathways influenced by localized hydrodynamic features such as shear flows or estuarine fronts. They also enable straightforward comparisons of the dynamics of the different types of particles and provide a simplified framework to take into account processes such as beaching, refloating, biofouling, and fragmentation (Sousa et al., 2021; Gorman et al., 2020; Shen et al., 2022; Elisei Schicchi et al., 2023; Sun et al., 2022). Backward-tracking modelling is another significant advantage, as discussed in Sun et al. (2022), often used to identify the potential sources of microplastic pollution. Additionally, the simulation of pathways also makes LNMs particularly suitable for assessing source-to-sink relationships, which are valuable for risk assessments and helping in management and mitigation strategies. However, LNMs have neglected some key physical processes, such as bottom deposition-resuspension or varying water density, which may particularly impact transport trends in systems with sharp density gradients. Nevertheless, LNMs can easily evolve to take into account these processes (Jalón-Rojas et al., 2024b) as discussed in section 2. In addition, LNMs need a high number of particles to get representative trends, but, as discussed in section 4, sensitivity tests may allow us to optimize it.

ENMs can, in turn, easily reckon with complex parameterizations of resuspension or vertical mixing as well as varying water density, as they are already considered in existing transport modules for sediment transport (Cancino and Neves, 1999). They can also easily incorporate interactions with fine sediments using state-of-the-art flocculation parameterizations used for sediment dynamics. ENMs may also facilitate analysing the influence of different hydrodynamic processes on transport dynamics by decomposing the momentum equation as in studies on suspended sediment transport (Xiao et al., 2020). However, simulating buoyant particles with rising velocities presents a challenge, as their upward movement requires specific parameterizations that are not always straightforward to integrate within traditional sediment transport frameworks. In contrast, LNMs are best suited for capturing buoyant microplastic behaviour, as they allow individual tracking of particles by updating dynamic and evolving properties.

Both ENMs and LNMs enable the repeated introduction of particles into different regions of the domain at various time intervals within a simulation. As ENMs focus on the concentration of microplastics, which changes over time, the repeated release can be modelled by adjusting the source term or boundary conditions at specific intervals. In LNMs, the repeated releases are more straightforward because each particle is injected at a specific time and then tracked. This feature allows for precise control and tracking of individual particle releases at specified times, facilitating a detailed examination of scenarios involving multiple release events and sources, as shown by Pilechi et al. (2022).

Calibration and validation of ENMs can be very challenging as they require high-quality observational data to compare with the simulated mass concentration of microplastics. Validation of LNMs is also challenging, but comparisons with observed trajectories from Lagrangian drifters can be more straightforward, providing validations at least for floating debris. Furthermore, ENMs typically require less computational time than LNMs, which can be particularly demanding in three-dimensional approaches when considering a wide range of processes. Nevertheless, the offline use of LNM models allows for simulating several scenarios in parallel without the need to re-run hydrodynamic simulations, thereby saving computational resources.

It should be noted that complex ENMs or LNMs can also be applied to simplified or idealized estuarine configurations to capture key processes and interactions. This "hybrid" approach avoids the limitations of exploratory models while enabling a simplified comparison of different scenarios or estuary configurations, including assessments over longer periods. However, identifying transport processes is less straightforward than exploratory methods, and simulations require more computational resources. For example, Shettigar et al. (2024) used an idealized estuarine configuration to compare the ENM of discrete classes of particles with the new PBE modelling approach, allowing for simplified comparisons of transport trends.

#### 3 Input data, model set-up, and model validation







Regardless of the selected approach and specific parameterizations, all the numerical models need careful preparation of input data and a systematic setting-up of the model framework before performing any simulation. Depending on the specific study objectives, the model setup can vary across different aspects such as domain size, grid dimensions, time resolutions, selected transport parameters, and microplastic characteristics. Ensuring an accurate representation of input data is crucial for capturing realistic conditions and producing meaningful predictions.

For microplastic transport studies, both hydrodynamic and transport models require comprehensive input data, meticulous set-up procedures, and rigorous calibration and validation using in situ observations to ensure the accuracy and reliability of the model prediction. The essential setup of the hydrodynamic modules involves (a) defining the geographical domain and constructing a computational mesh using bathymetric data; (b) initializing water levels, velocity fields and potentially salinity and temperature; (c) defining boundary conditions with meteorological and hydrological forcings; (d) setting up of model parameters such as bottom roughness or turbulence closure schemes. These parameters can be sourced from satellites, in-situ monitoring stations, and global ocean models. Readers can refer to Winterwerp et al. (2022) for detailed information on the parametrization of hydrodynamic modules in estuarine applications. Most hydrodynamic module configurations such as the computational grid, coastline, and bathymetry are also used in the transport module. Nevertheless, when reading hydrodynamic data offline in LNMs, these configurations can be modified or optimized for simulating microplastic transport (e.g., cutting the domain or interpolating outputs onto a new grid).

Specific parameters for the setup of transport modules include tracking parameters (e.g., calculation time step, simulation duration, number of particles or initial concentrations, release points) and transport process parameters (e.g.,  $K_h, K_v, W_{MP}$ ). In this section, we provide an analysis of the selected parameters and the validation processes used in the previously reviewed studies, which are summarized in Tables (1 and 2). A discussion and set of recommendations based on this analysis will be presented in Section 4.

# 3.1 Tracking parameters







The configuration of tracking parameters varied widely among the evaluated studies (Table 1 and 2), reflecting diverse approaches and objectives in modelling microplastic transport. The parameters, mainly applicable to numerical ENMs and LNMs, include:

- Release points. Defining the particle release positions (geographical position and depth) is an important step in ENM, LNM and PBE approaches, as it determines the initial distribution of microplastics. As highlighted in Tables 1 and 2, the choice varies across studies, mainly depending on the study focus or goal. For instance, the studies examining microplastic dispersion from wastewater effluents in an estuary located the release points near the wastewater treatment plants (Sousa et al., 2021; Shiravani et al., 2023). Conversely, when the goal is to use backward trajectories to identify the potential sources of microplastics, release points can be located in observed accumulation regions or near the system boundaries. For example, Sun et al. (2022) strategically identified the primary sources of microplastics entering the East China Sea through four main release points.
  - Particles can be released at a single point or distributed throughout the systems, either at a single moment or following temporal intervals (Defontaine et al., 2020; Gorman et al., 2020; Pilechi et al., 2022). In estuaries, releasing particles at different regions helps to understand the impact of different domain-specific physical processes, such as density-driven circulation, tidal straining or tidal asymmetry, among others, which vary spatially, on the trapping and export of microplastics to the ocean. In summary, our review indicates that there is no general rule for defining release points, as this depends on each application, the actual sources of microplastics, the simulation type (forward or backwards), and local environmental conditions. It seems challenging to account for all the potential sources present in an estuarine environment, probably due to the lack of data and computational time limitations.
- Number of particles and release frequency. Establishing a specific number of particles (Lagrangian) or initial particle concentration (Eulerian) is a fundamental yet challenging step, as it can influence not only the precision and reliability of the simulated results but also computational time. Depending on the application, an adequate number/concentration of particles at different sources can help provide a representative microplastic distribution. However, the lack of observations on microplastic concentrations and sources often makes it difficult to determine the most accurate values, potentially leading to uncertainties in the model results. As a result, many of the applications use exploratory values.
  - In the ENM approach by Defontaine et al. (2020), the model was initialised by releasing a patch of microplastics with a concentration of  $10 \ g/l^{-1}$  at a single time in the upper estuary. Shiravani et al. (2023) initialized the microplastic concentrations at locations representing the atmospheric flux and wastewater treatment plants, but did not mention the exact particle concentrations used for the initialization.

In previous LNM applications, the number of released particles can vary from a few to several thousands or millions based on the specific application and objectives of the study. For instance, the study by Elisei Schicchi et al. (2023) used only 4 particles to examine the influence of forcings like tide, waves, and wind in different environmental scenarios

in the Rio de La Plata estuary, which may limit the representativeness of the results. By contrast, Pilechi et al. (2022) released up to 250000 particles to analyse the performance of their LNM in simulating diffusion, and Sousa et al. (2021) continuously released 193000 particles for model validation and 400000 particles instantaneously in one of the scenarios. The latter implementation included the examination of tidal influence on microplastic emission, with particles released at four tidal conditions- at the beginning of the ebb and flood during both spring and neap periods. The study revealed that during the ebb phase, nearly a quarter of the released particles remained near the islands while some portion reached the open ocean, and a negligible amount was transported into the upper estuary. Furthermore, during the neap periods, even a low percentage of the released particles was observed to reach the open sea. In general, it remains challenging to establish initial particle count or concentrations based on observations, and simulation results should be interpreted with consideration of the chosen number or concentration of particles used in the simulation.






- Time step. Assigning an appropriate calculation time step is crucial, as it directly influences the accuracy in resolving the advection-diffusion equations in estuaries. The output timestep can also be important depending on the specificities of the systems. For example, an estuary dominated by tides needs output time steps lower than applications in the open ocean. The selection of the time step depends on the time scale of the physical processes to be investigated. As shown in Table 1 and 2, different studies have used various values of time steps for calculation and outputs, depending on specific requirements.

For instance, Shen et al. (2022) utilized 1 minute as the calculation time step for ensuring the model's accuracy. The calculation time step can be much shorter such as 0.1 seconds as found in the study by Pilechi et al. (2022) to record the influence of small-scale processes such as turbulence at high resolution. In the same study, a time step of 15 minutes was utilized for the sensitivity tests with various advection schemes. In the study by Sousa et al. (2021) in a mesotidal estuary, an output time step of 30 minutes was employed to show the influence of tidal currents in the transport of microplastics. Recommendations for selecting the time step are further discussed in 4.

- Time Duration. Similar to other tracking parameters, the choice of time duration depends on the specific goal and study domain, as there is no standard model configuration (Table 1 and 2). It can also depend on the underlying environmental processes of the estuary that need to be captured. For investigations examining the variability over the spring-neap tidal cycle, a minimum of 15 days is required to cover the full cycle, as done in Pilechi et al. (2022) and Sousa et al. (2021). Based on observations in the Yangtze estuary, Shen et al. (2022) chose one month for their simulation, as this timeframe corresponds to a biologically sensitive period when numerous species of birds visit the estuary to lay eggs and inhabit. Elisei Schicchi et al. (2023) also selected a simulation duration of one month to examine the influence of different forcings on microplastic transport. In another example, Sun et al. (2022) decided to implement a one-year backward simulation to fully understand the potential sources of microplastics in the East China Sea. Additionally, seasonal variations in the transport of microplastics can be studied by performing the simulation for at least one continuous year or by selecting different seasonal periods for the analysis (Gorman et al., 2020; Sun et al., 2022).

#### 3.2 **Transport process parameters**








The most common and important parameters to model microplastic transport processes are  $K_h$  and  $K_v$  for tracers or particles and  $W_{Mp}$ . These parameters rely on the domain environment, the particle properties, and the aim of the study. More complex applications can also include parameters related to deposition, re-suspension, beaching, refloating, flocculation or biofouling. However, this section focuses on the fundamental transport parameters related to mixing and particle dynamical properties used in previous studies. Unlike sediment transport models, the calibration of particle-related parameters such as  $W_{Mp}$  or flocculation/biofouling potential is challenging due to the variety of microplastics in the environment varying in size, shape and density, the scarcity of observational data, and the difficulty in acquiring such data (Defontaine et al., 2020; Jalón-Rojas et al., 2024a). Observations are therefore crucial for modelling studies as they help identify and characterize the main microplastic types to be modelled in the domain, ensuring realistic representations.

- Diffusivity coefficients. The modelling studies reviewed here (Tables 1 and 2) have assigned various values for  $K_h$  and  $K_v$ , reflecting different modelling approaches and domain-specific requirements. The choice of  $K_h$  and  $K_v$  in estuaries mainly depends on water turbulence driven by currents, wind, waves, and stratification in the water column. These parameters can be derived from observational studies, existing literature, or turbulence models (as outlined in Section 2).

As previously discussed in Section2, most studies have adopted constant values for  $K_h$  and assumed  $K_v$  to be constant too or, in some cases, equivalent to  $A_v$ . For example, in the idealized model study in Ría de Vigo, Díez-Minguito et al. (2020) used 0.0045  $m^2s^{-1}$  for  $K_v$  and no information about  $K_h$ . In a more realistic 3D model study in the same domain, Sousa et al. (2021) assigned values of 0.0001  $m^2s^{-1}$  and 5  $m^2s^{-1}$  for  $K_v$  and  $K_h$  respectively, highlighting the variability in choices for a same system, even when the values stay within the same order of magnitude. However, it should be noted that the latter study neglected the vertical movement of the particles. Elisei Schicchi et al. (2023) also allocated 0.0001  $m^2s^{-1}$  for  $K_v$  in the Rio de la Plata and a slightly smaller value of 1  $m^2s^{-1}$  for  $K_h$ . These three studies did not provide explicit justifications for the selected values, but the chosen values fall within the range found in the literature. Gorman et al. (2020) referred to values from 10 to 100  $m^2s^{-1}$  as typical for  $K_h$  in Santos Bay Estuary, citing previous studies in the domain, although the specific values utilized were not detailed.

In the Guadalquivir estuary,  $K_v$  was decomposed in a tidally-averaged component equal to 0.0123  $m^2s^{-1}$  based on previous studies, and a fluctuating component estimated from observations of vertical profiles of longitudinal current and salinity. Therefore, the lower estuary was characterised by the tidally-averaged value and the upper estuary, characterized by higher turbulence, by values up to  $0.19 m^2 s^{-1}$  (Bermúdez et al., 2021). In the Saint John estuary, Pilechi et al. (2022) compared microplastic transport patterns using constant  $K_h$  (1 and 10  $m^2s^{-1}$ ) and  $K_v$  (0.00001, 0.0001, 0.001, 0.01  $m^2s^{-1}$ ) values, along with time-space varying values based on the hydrodynamic model. They compared the real case scenario results and found that the one with time-space varying  $K_v$  is more accurate. ENMs adopting  $K_v$  equal to  $A_v$ from the hydrodynamic modules (Defontaine et al., 2020; Shiravani et al., 2023) did not specify the order of magnitude of these parameters.

From the above discussions, it is clear that different studies have adopted various diffusivity coefficient values. The challenges associated with selecting appropriate mixing parameters have been further discussed in Section 4, along with the recommendations.

- Terminal velocity. Another important step in model setup is deciding the types of particles to be modelled, i.e. their physical properties (size, density, shape) and consequently their settling or rising velocities. For example, denser, larger and more spherical particles generally sink faster, while lighter, smaller and irregularly shaped microplastics may rise or be transported as suspended load in the water column (Khatmullina and Isachenko, 2017; Kowalski et al., 2016; Al-Zawaidah et al., 2024). This is a challenging task due to the diverse amount of particles present in aquatic environments (Kooi and Koelmans, 2019). As previously outlined, external processes like aggregation or biofouling can further modify terminal velocities, leading suspended particles to be deposited and floating particles to be in suspension. Most of the evaluated studies assigned particle properties based on literature values for microplastics found in the environment, rather than focusing on the most abundant particle types specific to the study site, often due to the lack of observational data. The choice of particle properties can also depend on the study's specific goal.

The diversity in the choice of these properties is evident in the reviewed applications. For instance, Pilechi et al. (2022) tested different advection schemes and diffusions using neutrally buoyant particles with zero settling velocity. Sousa et al. (2021) examined microplastics with 4 size categories ( $10~\mu m$ , 1~mm, 2.5~mm and 5~mm) and different densities ranging from 900 to  $1020~kgm^{-3}$  to evaluate differences and similarities in their trajectories from wastewater treatment plants in the Ria de Vigo Estuary. By contrast, Díez-Minguito et al. (2020) considered pellets, fibres, and fragments with different densities and sizes, assigning  $W_{Mp}$  values of  $0.02~ms^{-1}$ ,  $0.0067~ms^{-1}$  and  $1.31\times10^{-7}ms^{-1}$ , respectively, based on state-of-the-art formulations to study microplastic distributions during upwelling-downwelling conditions in the same estuary. Similarly, Defontaine et al. (2020) evaluated the dispersion of three categories of microplastics with  $W_{Mp}$  values of  $4~mms^{-1}$ ,  $127~mms^{-1}$  and  $0~mms^{-1}$ , corresponding to sizes of 0.5~mm, 4.09~mm and 3~mm, respectively, with different densities. In the Garonne Tidal River, Jalón-Rojas et al. (2024a) used three  $W_{Mp}$  values to represent various microplastic categories found in aquatic environments and examined the impact of river discharge and tides on their trapping:  $0~mms^{-1}$  (neutral buoyancy, including microplastics with densities similar to water or biofouled lighter polymers),  $0.5~mms^{-1}$  (small microplastics with low settling velocities) and  $2~mms^{-1}$  (polyester microfibers, 1-5~mm, from fishing nets).

Three studies had the advantage of relying upon observations at their study sites to set up the particle selection. In the Yangtze estuary (Shen et al., 2022), fibrous particles of PE, PP and PVC with a length of 0.8 m and a diameter of 100  $\mu$ m were chosen for simulation based on pre-existing observations in the literature. Similarly, in the Guadalquivir estuary, observations revealed that PE was the dominant polymer type in the system. Based on it, Bermúdez et al. (2021) selected particles with this polymer and a representative rising velocity of 0.0046  $ms^{-1}$  corresponding to a bulk density of 980  $kgm^{-3}$  and an equivalent particle diameter of 2.3 mm (representing large microplastics 1-5 mm) according to

Waldschlaeger et al. (2020). Furthermore, Gorman et al. (2020) focused on floating pellet particle distribution, based on literature reviews of Santos Bay and the adjacent estuary.

It is important to note that incorporating all types of particles present in a system is challenging due to the difficulty in setting a wide range of microplastic properties, represented by diverse terminal velocities, which can vary spatially and temporally. For studies where considering a wide range of particles is a priority goal, the novel approach proposed by Shettigar et al. (2024) (Subsection 2.4) offers a promising tool.

# 3.3 Model validation and sensitivity analysis








Model validation is a critical step in the development and application of microplastic transport models, ensuring their accuracy and reliability in representing real-world processes. Strategies for model validation include hydrodynamic validation and Lagrangian validation. Hydrodynamic validation typically adopts an Eulerian perspective to ensure an accurate representation of the physical environment and driving processes. Lagrangian validation involves comparing simulated and observed trajectories of drifters. A third strategy consists of the direct comparison of simulated and observed microplastic concentration collected from water samples, akin to sediment dynamics. This approach is challenging due to uncertainties in initial particle distributions, the difficulty in capturing complex processes such as flocculation and biofouling and sampling limitations, among others (details in Section 4).

In estuarine or coastal systems, where the circulation is affected by complex interactions between tides, river flow, bathymetry, residual flows, and stratification, must be realistically captured to estimate the transport patterns and potential trapping zones of microplastics. Therefore, hydrodynamic modelling and validation are not only the first step but also the most controlling factor in determining the realistic transport mechanisms. For validation of the hydrodynamic module, field measurements such as water level, current velocities, salinity, and temperature are used both to calibrate model parameters like bottom roughness and to validate predictions. Hydrodynamic validation information is often scarce in microplastic modelling studies at oceanic or regional scales, as they typically rely on operational hydrodynamic models. In contrast, all the studies discussed here have performed hydrodynamic validations with a few validating during the study period (Shiravani et al., 2023; Shen et al., 2022; Sun et al., 2022), and the majority relying on previously calibrated hydrodynamic models for their study location (Defontaine et al., 2020; Sousa et al., 2021; Pilechi et al., 2022; Elisei Schicchi et al., 2023). In estuarine systems, validation during the study period might be particularly relevant, as seasonal and interannual changes in morphology and sedimentary environments may significantly influence hydrodynamics (Jalón-Rojas et al., 2018, 2021). Validating other parameters can be necessary depending on the study goal. For example, Shiravani et al. (2023) and Shen et al. (2022) validated suspended particulate matter as their research focused on the interaction of microplastics with fine sediments.

One of the major challenges in microplastic transport modelling is the lack of robust validation of the transport module, primarily due to limited observational data, difficulties in sampling in dynamic estuarine environments (Defontaine and Jalón-Rojas, 2023), and the complexities of reproducing realistic conditions, as discussed above. However, most of the reviewed studies (Tables 1 and 2) endeavoured to provide some form of validation using available datasets from field campaigns or previous research and providing statistical metrics such as Pearson Correlation coefficient, Skill scores and Mean absolute

error (Shiravani et al., 2023; Sun et al., 2022; Shen et al., 2022; Gorman et al., 2020; Sousa et al., 2021). Sometimes, these validations require the conversion of mass concentrations into the number of particles, as done in Shiravani et al. (2023) and Pilechi et al. (2022), or vice versa, as done in Sousa et al. (2021). As suggested in Shiravani et al. (2023), the use of conversion formulas can introduce uncertainties and influence the validation processes.

Model simplifications, such as overlooking some transport processes, sources, or particle types, can particularly affect the validation processes. For example, in the Yangtze estuary study by Shen et al. (2022), validation improved when beaching was applied, reflecting the high beaching rate in the estuary. However, the authors also reported discrepancies at certain locations, which they attributed to using a constant  $K_v$  that does not capture spatial variability. Additionally, studies like (Sousa et al., 2021; Shiravani et al., 2023; Shen et al., 2022) highlighted that the omission of certain potential sources of microplastics in their study domain (e.g., rivers, cities, all existing wastewater treatment plants) contributed to discrepancies and poor validation at specific stations. By focusing on a subset of particles, models may overlook the contribution of less abundant or smaller-sized microplastics, which may significantly influence transport dynamics and accumulation zones.

Sensitivity tests are a useful tool in microplastic transport modelling for calibrating and optimizing models and testing scenarios. Given the uncertainties in parameterizations and validation, sensitivity tests help identify the most influential parameters, assess model robustness, and provide insight into the range of possible outcomes under varying conditions. Only a few studies discussed here have conducted sensitivity analyses to identify key parameters and processes and to explore the transport dynamics of different types of microplastics. For instance, Pilechi et al. (2022) conducted several sensitivity tests at the Saint John estuary, exploring the importance of selecting appropriate advection schemes and diffusion approaches to improve model accuracy. Most of the reviewed studies, however, focused on scenario tests with different forcings or particle types, without addressing factors like computational time step, grid resolution, diffusivity coefficients, or number of released particles. Moreover, while published studies have proposed a series of specific dominant explanations for the discrepancies between models and observations, we can expect that many, if not all, of the mentioned issues can affect the model validation and calibration, being a matter of quantity, quality and relevancy of observations, robustness of the comparison methods and performance and representativeness of both hydrodynamic and transport models. In the next section, we outline recommendations to enhance the implementation and utility of sensitivity analyses in future studies.

#### 4 Challenges and recommendations






Modelling studies of microplastic transport face several inherent challenges, many highlighted in the previous sections, that complicate the development, setup, and validation of microplastic transport models. This section outlines these challenges and provides recommendations and good practices to enhance the accuracy and reliability of the models. The base recommendations, particularly concerning model construction and setup, are intended for a general scope, i.e. already well understood by modellers in the fields of oceanography or riverine hydraulics, but will remain of interest for newcomers discovering the complexity of modelling the transport of microplastics in estuaries. The main challenges and recommendations are summarized in Figure 3.

- Model construction and setup. Constructing a process-based model involves several complex steps, particularly in setting up the grid, the bathymetry, and the initial and boundary conditions. Selecting between structured and unstructured grids requires careful consideration, especially when dealing with complex geometries. Creating effective model grids in such contexts is inherently challenging. Additionally, balancing the need for high-resolution grids with computational constraints demands strategic planning. Acquiring high-resolution bathymetry data, essential for model accuracy, is often logistically difficult and expensive. Similarly, defining reliable initial and boundary conditions over appropriate temporal and spatial scales is crucial but equally challenging due to data gaps.







Recommendations: As highlighted by Winterwerp et al. (2022) for hydro-sedimentary models, a first step for model construction consists of a clear definition of the study objectives and research questions, along with the development of a conceptual model to frame and guide decision on key aspects; model domains, targeted spatial and temporal scales, simulation durations, relevant transport processes (Fig. 1), transport modelling approach, and parameterizations. For example, if analyzing floating particles and source-to-sink patterns is identified as a priority in this first step, LNM would be the most suitable choice. This step also helps to pinpoint data gaps and plan targeted field campaigns to gather essential observations for model setup (e.g., bathymetry, input data) and validation (e.g., current velocities). Given that these measurements are generally expensive and cannot cover the entire domain, system understanding and a well-defined conceptual model are essential for designing an efficient and cost-effective measurement program. A key recommendation is to prioritize high-quality bathymetric data, as hydrodynamic model outputs, particularly current velocities in shallow waters, are highly sensitive to bathymetry (Winterwerp et al., 2022). Furthermore, accurately representing bottom roughness is crucial for hydrodynamic modelling. In estuarine environments, this may require accounting for spatiotemporal variations driven by seasonal changes in the sedimentary environment (Jalón-Rojas et al., 2021; Do et al., 2024). This first step is also critical for defining the key forcings of the hydrodynamic model and determining the degree of complexity. Models should be sufficiently complex to capture the relevant physics for the study objectives, but unnecessary complexity can slow simulations, make results harder to interpret, and sometimes even reduce accuracy (Winterwerp et al., 2022).

Selection of transport processes and their parameterizations. An important and particularly challenging step in transport model setup is the parameterization of transport processes and the choice of parameter values. The dynamic nature and complex hydro-sedimentary processes of estuarine systems often necessitate the inclusion of processes or parameterizations not typically considered in microplastic transport studies at regional- or oceanic scales, such as bottom resuspension, water density effects on terminal velocities, spatio-temporal variability of vertical mixing, or beaching/refloating. This, combined with the potential need to account for microplastic-specific processes, such as aggregation, degradation, flocculation, biofouling, and fragmentation, can significantly increase the complexity of the model configuration and computational demands. Considering that numerous processes can also multiply the number of calibration parameters, potentially amplifying uncertainties in the model configuration and results. Selecting key processes and adapting parameterizations to align with the study objectives is, therefore, a crucial step in ensuring model relevance.

accuracy, and efficiency, but the diversity of parameterizations, as reviewed in this study (Section 2), can make decisions challenging.

**Recommendations:** Selecting and incorporating relevant processes while avoiding unnecessary complexity requires a comprehensive understanding of the region, including its hydrodynamics as well as its physical, chemical and biological conditions. For instance, considering the effect of varying water density on microplastic terminal velocities may be more important in stratified estuaries than in well-mixed estuaries. Similarly, flocculation processes can be particularly relevant for small microplastics in hyperturbid estuaries, while biofouling can be more significant for larger microplastics in systems characterized by high biological productivity and organic matter availability. Therefore, conducting a comprehensive literature review is always an essential step in the modelling process.

Literature review can also help to select the most adapted parameterization for the study objectives. For example, selecting the most appropriate formulation for calculating particle terminal velocities from the wide range of available options (Tables 1 and 2): large microplastics require specific formulations that account for their shape (Waldschlager and Schuttrumpf, 2019a; Jalón-Rojas et al., 2022), whereas classical formulations developed for natural particles are well-suited for small microplastics (Dittmar et al., 2024). In addition to the selection and parameterization of particle-related processes, the accuracy of the numerical methods is of primary importance for transport modelling. The fourth-order RK method has been recommended due to its balance of accuracy and stability (Pilechi et al., 2022). Real-field data further helps to pinpoint dominant processes that should be included in simulations and for estimating appropriate parameter values. Lagrangian drifter measurements can help estimate horizontal dispersion coefficients and reduce uncertainty.

We recommend performing various sensitivity tests before actual simulations to identify key processes and parameters influencing transport patterns and to optimize the model configuration, reducing computational time. In LNMs, the most commonly used model approach (Table 2), sensitivity tests can be done to optimize grid resolution (in offline coupling), calculation time step, and the number of released particles, helping to identify the balance between accuracy and computational resources. Like any other numerical model, determining the grid resolution and time steps requires adherence to the Courant-Friedrichs-Lewy (CFL) criterion. Therefore, it can be recommended to choose a spatial grid size fine enough to capture the physical processes and a maximum allowable time step that ensures the CFL condition. Preliminary sensitivity tests can also help evaluate the uncertainty associated with unknown parameters and identify the relevance of factors such as spatiotemporally varying diffusivity coefficients, buoyant effects caused by the effect of varying water density or bed roughness effects on microplastic resuspension caused by spatially varying sediments.

Selection of particle characteristics. As highlighted in previous Sections, the vast variability in the shape, size, density, and surface properties of environmental microplastics, all of which influence how particles interact with water, sediment, and biological components, presents an important challenge for model configuration. It is necessary to strike a balance between selecting key particle characteristics based on the specific objectives of the study and the feasibility of running multiple simulations.

Recommendations: The best way to deal with this challenge is to gather comprehensive and region-specific data on the characteristics of the most common microplastics present in the modelled system. Combining literature review, field observations using standard protocols for sampling and taking into account estuarine spatial-temporal variability scales (Defontaine and Jalón-Rojas, 2023), and laboratory analysis for determining the particle characteristics would be the perfect strategy. However, such analyses imply substantial financial and time costs and are not always feasible. A general recommendation would be to focus on the most common particle types present in these systems. Landebrit et al. (2024) found that small floating microplastics predominate in surface waters of European rivers and estuaries while settling, but mobile fibres dominate in the water column of several coastal systems (Lefebvre et al., 2023; Bagaev et al., 2017). Selecting two particle categories—floating and small settling particles—can be a practical solution for exploring different behaviours. Nevertheless, Jalón-Rojas et al. (2024a) demonstrated that trapping processes in the Garonne tidal river were very sensitive to settling velocities, even when particles represent similar types, suggesting that results should be interpreted with caution when using single terminal velocity values.

Selecting ENMs allows for the simultaneous definition of several microplastic classes, each with distinct characteristics that can enable a more comprehensive representation of particle behaviours. In LNMs, a wide spectrum of settling velocities can be simulated by distributing them across a large number of individually released particles, capturing the variability within and between microplastic classes. This strategy was employed in an ocean-scale application by (Pierard et al., 2024a), where each particle was initially assigned a size randomly drawn from a uniform distribution within a specified range, and settling velocities were calculated accordingly. Data on microplastic properties from local sources can further refine these release strategies by providing more accurate input on particle size and density distributions.

- Selection of release points and the number/concentration of particles. Microplastic transport models require accurate release point locations, which are typically based on real-world conditions. As shown in our review, these release points mainly include rivers, cities, and water treatment plants (Table 1 and 2). Our review also highlights that one of the most challenging steps in the microplastic transport model setup - arguably the most challenging - is determining temporal variability, intensity and frequency of the releases due to the lack of data. None of the previous studies were able to represent "realistic" release conditions fully. Furthermore, representing diffusive sources like runoff or atmospheric input is a difficult task as these sources are spatially and temporally varying, often lacking precise data on their magnitude and distribution. The inaccuracy in the release point locations may lead to errors in predicting microplastic transport patterns.

**Recommendations:** Using available data to approximate release conditions would be the ideal strategy. Where local data is unavailable, other methods can be implemented. Relying on regional studies, global datasets, or model estimations could provide useful approximations, for example, for river inputs. Given the challenges of obtaining precise temporal data on release intensity and frequency, modellers can incorporate realistic assumptions based on known seasonal patterns or surrogate data, and use sensitivity analyses to test how different release scenarios affect model outcomes. Empirical models, such as Weiss et al. (2021), which incorporate parameters such as population density and river flow

intensity, might be particularly effective for estimating riverine microplastic inputs by linking these factors to expected emission rates.

Additionally, modellers can focus on capturing broad trends in microplastic transport rather than focusing on exact precision, ensuring that model outputs remain relevant and robust despite uncertainties in input data. In LNMs, resulting microplastic distributions can be quantified as the number of particles relative to the released ones using probability density maps (Jalón-Rojas et al., 2019b; Hatzonikolakis et al., 2022). In that case, the number of released particles should be high enough to provide statistically robust results while remaining computationally feasible. This balance can be achieved through sensitivity that compares dispersion trends across simulations with varying particle counts, allowing for the selection of an optimal number of particles without compromising results accuracy. In ENMs, initial particle concentration can also be determined through sensitivity tests. However, as previously mentioned, high concentrations of particles are needed to avoid diffusion problems.

**Figure 3.** Recommendations for microplastic modelling in estuaries




- Model calibration and validation. Calibrating and validating microplastic transport models are challenging processes due to all the previously mentioned uncertainties in input data and model assumptions. As highlighted in our review of validation strategies in previous studies (Section 3.3), one major issue is the lack of high-resolution, long-term observational data on microplastic concentrations, similar to the extensive datasets available for suspended sediment, which

is essential for robust model calibration (Do et al., 2024). The variability in microplastic types, particle behaviour, and environmental conditions further complicate the process, as models may need to account for a wide range of release scenarios and hydrodynamic conditions. Additionally, validating model outputs against in-situ measurements is difficult, as monitoring efforts often lack consistency, spatial coverage, and temporal resolution (see review by Defontaine and Jalón-Rojas (2023)), making it hard to match model predictions with observed distributions.

Recommendations: Given the crucial role of current velocities in driving microplastic transport, a key recommendation would be to ensure a thorough validation of the hydrodynamic model using high-quality, site-specific data on current velocities and water level, alongside statistical metrics such as RMSE, bias, and correlation. The validation of salinity can also serve as a useful indicator for verifying the accuracy of diffusion patterns, particularly when the microplastic transport module applies diffusivity coefficients derived from the hydrodynamic model. Moreover, discrepancies in salinity validation may highlight potential issues with numerical diffusivity. As pointed out in Winterwerp et al. (2022), a good practice is to use data from different periods for the calibration and validation phases to enhance the robustness and reliability of the model. Incorporating drifter or dye data for validation of the transport model can enhance confidence in simulations, particularly in complex environments like estuaries. When in-situ measurements of microplastic concentrations are available, qualitative comparisons can be proposed to identify general trends rather than achieve precise quantitative agreement in both Eulerian and Lagrangian models. For instance, this approach has been applied in an application in the Mediterranean Sea (Baudena et al., 2022), providing insights into the reasonably accurate estimation of broad patterns of microplastic distribution and transport. In estuarine applications, this qualitative comparison can be performed by examining the broad patterns of variability in microplastic concentrations over key time scales, such as tidal ranges, spring-neap cycles, or seasonal changes.

# 5 Future directions







The above section highlighted the importance of further exploration of several critical aspects to improve the accuracy and applicability of microplastic modelling in estuaries. Here, we outline potential future directions that could significantly advance the field.

Accurate setup and validation of microplastic transport models are highly dependent on robust field measurements. Future studies should prioritize comprehensive data collection to validate hydrodynamics, enhance the model parametrizations and inputs, and validate microplastic transport trends at least qualitatively. Improving validation methodologies is essential for establishing model credibility. Strategies such as incorporating diverse data sources, cross-validation techniques, or using independent reference standards can increase confidence in model predictions. To improve the reliability of the accompanied hydrodynamic models, future efforts should also focus on addressing the equifinality problem -the phenomenon where multiple set of parameters yield similar model outputs (van Maren and Cronin, 2016). For example, Do et al. (2024) minimized the calibration parameters by using abundant observational data, enhancing model robustness and reducing uncertainties.

A promising future direction is the application of data assimilation techniques when field data is available, which have begun to be used in oceanic-scale microplastic modelling (Peytavin et al., 2021). This strategy can reduce uncertainties and biases, improving the representation of the mean state. Performing sensitivity tests on multiple release scenarios and calibration parameters may help to capture the improved mean state from data assimilation (Pierard et al., 2024b). However, comprehensive datasets are still largely unavailable due to challenges in collection (Defontaine and Jalón-Rojas, 2023), but advancements in monitoring technologies (Ruiz-Gonzalez et al., 2024) and collaborative efforts offer promising opportunities for improvement.

Transport models should initially aim for an accurate representation of transport patterns through advection-diffusion and vertical transport processes. Once these foundations are established, more advanced and complex mechanisms, including floculation, fragmentation, and biofilm development, can be incorporated into the simulations. Future modelling studies will benefit from recent and ongoing laboratory experiments that will advance the parametrization of these processes (Poulain-Zarcos et al. (2024); Wu et al. (2024b), Ruiz-Gonzalez et al. (in prep)), ensuring that the model better represents actual physical, chemical, and biological processes in the system and improves predictive capabilities.

A key future direction involves connecting estuaries with rivers, adjacent coastal systems, and the continental shelf in models to explore the continuum of microplastic transport across these interconnected environments. This would require accounting for often-overlooked processes such as wave processes Jalón-Rojas et al. (2024b) and wave-current interactions, which can play a crucial role in microplastic transport. Such an integrated approach would provide a more holistic understanding of microplastic dynamics, capturing the exchanges between freshwater, estuarine, and marine systems and offering insights into the fate and pathways of microplastics at regional and global scales. Furthermore, extending the temporal scope of the studies will help to capture the long-term trends of the microplastic transport, trapping and deposition within the system, an important research focus in the context of global change. However, this is challenging due to significant interannual changes in the estuarine environment (e.g. bathymetric changes), which complicate efforts to model and predict transport processes reliably over extended periods.

Adding multiple transport processes and targeting longer time scales means increasing the complexity of the model, which may present computational constraints. As underscored in the previous section, sensitivity tests are a helpful tool to assess the relative importance of processes influencing microplastic transport within the domain. Identifying the key drivers enables targeted model refinements that balance realistic simulations with optimised computational efficiency. Advanced computational techniques, such as machine learning, are promising approaches to reduce computational demands. For instance, Fajardo-Urbina et al. (2024) recently combined LNMs and deep learning techniques to predict particle paths in coastal environments, demonstrating enhanced computational efficiency and improved predictions.

Finally, fostering interdisciplinary collaboration between oceanographers, engineers, ecologists and chemists is essential for the comprehensive development of microplastic transport models. These partnerships enable the integration of diverse expertise, from hydrodynamics and particle dynamics to chemical degradation and ecological impacts, developing models that more holistically address the complexity of microplastic dynamics and distribution in estuarine and coastal environments.

By addressing these directions, future research can significantly improve the reliability, scope, and applicability of microplastic transport models, paving the way for more effective strategies to mitigate the environmental and ecological impacts of microplastic pollution and inform policy decisions and management practice.

Code availability. No code was used in the preparation of this review. No data were processed in the preparation of this review. This review relies solely on the methodologies presented in previously published studies, which are detailed in Tables 1 and 2.

Author contributions. **BJK**: Formal analysis, Investigation, Data Curation, Methodology, Writing - Original Draft.**IJR**: Conceptualization, Methodology, Visualization, Supervision, Writing - Review and Editing, Funding acquisition. **DS**: Methodology, Supervision, Writing - Review and Editing.

Competing interests. No competing interests

Acknowledgements. This work was supported by the French National Research Agency (ANR) through the project PLASTINEST (ANR-22-CE01-0011-01). We gratefully acknowledge the constructive and insightful comments provided by the two anonymous reviewers, which have significantly contributed to improving the quality and clarity of this manuscript.

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
