# Peer review of "Modelling Microplastic Dynamics in Estuaries: A Comprehensive Review, Challenges and Recommendations"

_EGUsphere, 2025_

## Author Response (AR1)

We sincerely thank the two anonymous reviewers and the community commenter-Nithin Achutha Shettigar for their constructive and insightful comments and valuable suggestions. Their comments have helped us improve the clarity and quality of the manuscript. Below, we outlined detailed responses to each comment.

Community Comment

line 450 - internal coordinate  $\eta$  representing the particle size. Instead, it is to be  $\xi$

We corrected the symbol.

Reviewer 1

I am delighted to read this thorough and insightful review of microplastic dynamics modeling in estuaries. The authors have done an exceptional job synthesizing the current state of research, addressing key challenges, and providing well-considered recommendations for advancing this critical field. Their comprehensive approach not only highlights the complexities of microplastic transport and fate in estuarine environments but also offers valuable guidance for future studies, making this a significant contribution to environmental science.

The characteristics of microplastics are known to influence transport modeling. However, I believe the modeling of ocean currents is the most critical factor. Although the authors acknowledge this in the manuscript, I suggest emphasizing this point further.

We thank the reviewer for the positive comments on our manuscript. Regarding the imporance of modeling accurately ocean and estuarine currents, other than highlighting this point in the section "Challenges and Recommendations", we have further emphasized it in lines 693-696: "In estuarine or coastal systems, where the circulation is affected by complex interactions between tides, river flow, bathymetry, residual flows, and stratification, must be realistically captured to estimate the transport patterns and potential trapping zones of microplastics. Therefore, hydrodynamic modelling and validation are not only the first step but also the most controlling factor in determining the realistic transport mechanisms."

**Reviewer 2**

This manuscript surveys the literature on modelling of microplastic transport in estuaries. The manuscript categorizes studies according to their approaches for model setup and identifies important aspects of model implementation and parameter selection. The authors make recommendations for key considerations in designing models for microplastic transport in estuaries and point out areas in need of additional research.

This review is timely given the increasing interest in predicting the fate of microplastics in the aquatic environment in general, and specifically in estuaries. There have been numerous studies along these lines in recent years to draw on. Bounding the topic of the review and providing the appropriate level of detail on the studies are always key considerations. The stated aim is focused in methodology (modeling), subject matter (microplastics), and geographic scope (estuaries). The challenge is to synthesize a representative range of studies and provide analysis that identifies strengths, weaknesses, and gaps in the field.

We thank the reviewer for the positive comments and thorough review of our

**manuscript, which helped us improve its quality.**

My major comment is that this review is weighted too heavily toward listing the details of what other studies of done and does not provide enough synthesis that would add value beyond the references to the individual studies. The discussion also wanders into related but distinctly different topic areas (such as hydrodynamic modeling) and provides excessive levels of details in equations and parameter choices that can easily be found in the citations, other topical reviews, or textbooks. This results in an overly long manuscript that risks losing reader interest before the main discussion on advantages and disadvantages of different approaches and recommendations for future steps. Trimming and reorganizing the text as suggested in the comments below would make for a more accessible review. The review has good quality content and appropriate scope, and some reorganization and streamlining of the text could greatly enhance its impact.

We agree that summarizing some details such as those related to hydrodynamic models and equations may improve the clarity and accessibility of the manuscript. We have made multiple modifications in that sense and deleted several general equations while adding more elements of synthesis throughout the text, many of which were inspired by the reviewer's helpful suggestions, for which we are grateful. At the same time, we would like to emphasize that this review is intentionally aimed at a broad audience interested in microplastics, from master's students to researchers from diverse disciplines, many of whom may not be familiar with modelling approaches and require sufficient contextual detail to understand the differences between existing methods and parameterizations. We believe that a certain level of technical elaboration (e.g. the introduction of key parameters and parameterization choices) is valuable not only for beginners and non-specialists engaging with modelling literature but also for more specialized readers, given the rapid evolution of this field and the lack of consolidated references. In conclusion, we aimed to balance technical detail with synthesis to support both non-specialists and experts. We maintained contextual depth while ensuring the content remained engaging and coherent. We have revised our text to improve readability while maintaining a clear progression from an overview of the modelling choices made in the reviewed studies to a discussion of key challenges, followed by recommendations and future directions based on the main findings. We discussed these elements with the Editor, who emphasized the importance of clearly identifying the recommendations and perspectives in the manuscript.

Line 16: missing "by" after "research"

**Done**

Line 19: "both" is used twice in the sentence and is only needed once.

**Removed**

Line 22: It's not clear what "privileged" means here.

**Changed the wording from "privileged roads" to "primary pathways"**

Line 85: The topical organization here is confusing. Four categories of models are identified: Eulerian analytical or idealized, realistic Eulerian, realistic Lagrangian, and Population Balance Equation. All of them rely on interfacing with a hydrodynamic model that represents the transport, mixing, and ambient conditions the plastic particles are experiencing, which is explained later in the text but still ambiguous at this point. The estuary being simulated could be idealized or realistic. The hydrodynamic model can solve the transport equations analytically (requiring simplifying assumptions) or numerically. The representation of the particles can be in Eulerian as concentration or Lagrangian as discrete particles. The distribution of particles sizes and characteristics can be done in numerous ways, including a single size class, multiple separate size classes, or multiple interacting size classes, which is what the Population Balance approach sounds like. Particle characteristics such as settling, resuspension, degradation, and biofouling can be represented at various levels of complexity. It would be more logical and informative to identify these distinct dimensions of model characteristics and discuss the tradeoffs along each dimension independently rather than grouping them more arbitrarily, as is currently the case. For example, idealized estuary models can use Lagrangian particle tracking to represent plastic transport (e.g., Bo et al. 2024, https://doi.org/10.1073/pnas.2401498121), but that approach does not fit in the organizational framework here. Similarly, it seems like the Population Balanced approach could be applied in a realistic or idealized hydrodynamic models, so it overlaps with other categories. More thoughtful organization of the orthogonal model attributes would help readers organize the topic and help guide more coherent synthesis of the studies.

We agree that modelling approaches can be classified according to different criteria (e.g., analytical vs. numerical; Eulerian vs. Lagrangian; idealized vs. realistic), and that our initial distinction—particularly labelling 2DV Eulerian models as "idealized"—may be confusing. While this terminology is common in hydro-sedimentary modelling, we acknowledge that other approaches can also be applied to idealized or academic configurations, as already noted in lines 518-523. In the revised version, we have clarified the classification scheme and introduced the four main types of transport models used in the literature, each based on different particle transport equations: Eulerian (semi-)analytical 2D vertical models (EA2DV), Eulerian numerical model, Lagrangian numerical model(LNM), and Population Balance Equations (PBE) (lines 88-89, 95, in the abstract and in the tables). We anticipated that both Eulerian and Lagrangian approaches can be applied to realistic or idealized setups, while the semianalytical and current PBE frameworks are limited to idealized configuration although future developments in PBE may enable more realistic applications (lines 522-523, 447). We also better explained the particularity of the PBE approach in using a continuous distribution of particle size classes (lines 426-432). To avoid excessive technical detail, we maintained representative equations that we believe clarify the differences between the four modelling strategies for a broad readership, while deleting other complementary equations as suggested by the reviewer in other comments.

Line 98: should be "Tables 1 and 2 compile and compare"

**We rewrote the typos**

Line 143: Since this is a review, it would be reasonable to leave out the details of model applications such as the governing equations and direct readers to the references cited. That might help streamline the text for a manuscript that is quite long.

We agree with the reviewer and have removed most of the classical equations (old Eqs. 4–6, 8–12, 14–15, 21–27), retaining only the general equations specific to each approach and those derived for particular cases, to highlight their distinct features,

**especially for readers less familiar with these methods.**

Line 162: Is the "FE" here the same as "E" in equation (5) and in the next paragraph? If so, please make them consistent.

**We removed the equation**

Line 165: This section seems to be interchanging sediment transport and plastics transport, which is confusing. There are analogies between the topics, but there can also be big differences. For example, the statement here is that the parameters are determined based on morphodynamic equilibrium, but that seems irrelevant to plastics transport. Suggest keeping the focus here on how sediment transport approaches can be adapted to plastics rather than getting into the details of the sediment transport equations.

We agree with the reviewer and added a part as a transition from sediment transport to microplastic transport in the revised manuscript in line 177-181 as "While this equilibrium concept originates from sediment dynamics, it can be adapted to microplastic transport by treating the system as quasi-steady and focusing on the net fluxes over a tidal cycle. Although microplastics do not contribute to bed transformations, adapting this approach offers a useful framework for quantifying net accumulation zones and transport pathways under tidally averaged conditions."

Line 190: Maybe this is coming later, but I expected more of a synthesis of the studies using the approach described in this section – key findings, sources of uncertainty, benefits and limitations. Identifying the approaches used is an important start, but it would be valuable to assimilate the diverse sources into coherent guidance for readers.

As noted by the reviewer, the synthesis of benefits, limitations, challenges, and recommendations from key findings is provided later in the manuscript, in dedicated sections 2.5 and 4. To guide the reader, we have added sentences in lines 107-108, 248, 600, 645-647, 692 and 735-736 indicating that these elements will be addressed in the upcoming sections. While we have incorporated some of these aspects earlier in the manuscript, this has been done to a limited extent to avoid repetition and to keep the manuscript as concise as possible.

Line 205: Similar to the comment above, the focus here should be on the microplastic aspects of the modelling rather than the hydrodynamics. Textbooks have been written about the tradeoffs between finite difference and finite element, or sigma vs z-coordinates, and it would be a disservice to try to cover those topics here. Instead provide context on the choices that go into using circulation models to simulate plastic transport, either with Eulerian or Lagrangian representation of the plastics.

We agree with the reviewer and have removed the detailed description of the discretization methods used for hydrodynamics. This section is now summarized in line 216 as "ENMs can be employed with several discretisation methods and grid structures based on the domain and the application with a citation. However, since the choice of vertical coordinate is important for simulating vertical microplastic transport, we have retained the explanation of this aspect in the text."

Line 233: Is the sense of the inequality in equation (9) correct? It seems to be saying that

deposition only happens for stress greater than the threshold rather than less than the threshold, which is counterintuitive. Similar for equation (8). Or consider removing these equations since they are standard for sediment transport modeling and can be found in the references.

**Removed the equation**

Line 239: Is this "E" from Equation (10) the same as that in Equation (5)? If not, change the nomenclature so they are not confused.

**Removed the equation**

Line 243: "Traditionally, modelling efforts are focused mostly on the floating particles leading to a stronger emphasis on the approaches suited for surface transport." This is confusing following the previous paragraphs, since they presume that particles are negatively buoyant and act like sediment. It would be helpful to discuss the role of buoyancy for plastics earlier on in the manuscript, and how that influences modeling choices.

We have modifed the text to avoid confusion by removing this sentence and revising line 245 as "ENMs are primarily employed to analyse the particle dynamics in the water column, typically applied to neutrally buoyant or non-buoyant particles" and also highlighted in Section 2.5 the choice of model can depend on the type of particle from line 501-505 as "ENMs may also facilitate analysing the influence of different hydrodynamic processes on transport dynamics by decomposing the momentum equation as in studies on suspended sediment transport" with a citation. "However, simulating buoyant particles with rising velocities presents a challenge, as their upward movement requires specific parameterizations that are not always straightforward to integrate within traditional sediment transport frameworks. In contrast, LNMs are best suited for capturing buoyant microplastic behaviour, as they allow individual tracking of particles by updating dynamic and evolving properties."

Line 245: "Furthermore, transport processes such as beaching, which involves the interaction of the particles with coastlines, cannot be effectively captured using ERMs, due to their fixed spacial framework." This is not true, as ERMs with wetting and drying would be able to deposit particles (sediment or plastics) in intertidal regions such as beaches. Please clarify.

We revised the text in response to the reviewer's comment by removing the original sentence and adding clearer explanations in lines 246–248 as "Fine transport processes with coastlines such as beaching have generally not been accounted for up to now, while potentially implementable with fine spatial resolution and accurate representation of the wetting and drying dynamics."

Line 247:How is this "lower preference for ERMs" identified? By less prevalence of Eulerian applications compared to Lagrangian? It's not obvious why practitioners would gravitate toward one approach or the other simply because of popularity...perhaps there are additional factors such as ease of implementation or efficiency that favor Lagrangian approaches?

We agree with the reviewer and have removed the confusing sentence "Additionally, the widespread adoption of the Lagrangian approach in global-scale plastic transport models has significantly influenced a lower preference for ENMs in subsequent plastic transport studies." replacing it with a clearer version in line 248-250: "Several factors discussed in Section 2.5 - such as the challenges in representing floating particle dynamics within Eulerian frameworks, their inability to explicitly resolve source-to-sink pathways, and the widespread adoption of Lagrangian approaches in global-scale plastic transport studies - may have limited the utilisation of ENMs in the study of microplastics."

Line 250: Not sure what "precedent" means here...suggest removing.

**Removed**

Line 258: Similar to a comment on the previous section, this section goes into unnecessary levels of detail on what the previous studies did (all of which can be found in the citations) and provides little synthesis on what the studies learned or what we can learn by comparing and contrasting them.

As argued above, this section is dedicated to reviewing the state of the art and therefore we discussed only what the previous studies have done; synthesis is provided in Section 2.5 "Advantages and limitations", Section 4 "Challenges and recommendations" and Section 5 "Future Directions", based on key findings from these studies. We also revised this section by removing the equations and striking a better balance, retaining essential information.

Line 270: Presumably the Dietrich (1982) settling velocity formulation was developed for sediment rather than microplastic, as is suggested here. Please clarify.

We modified the sentence in lines 267-268 as "In Shiravani et al. (2023), WMp was computed using the formulation proposed by Kooi et al. (2017) according to (Dietrich, 1982), a formulation developed for natural particles."

Line 355: Is this advection scheme analysis specific to plastics or more general for LPTMs? If the latter, that is getting away from the focus of this review. If the former, please add context on what makes the choice of advection scheme specific to or particularly important for plastics.

The study by Pilechi et al al. (2022) is specific to microplastics in estuaries and for that reason we have included it in our review (Tables 1-2). However, it can still be valid more generally for LPTMS. We modified the sentence to highlight that the study is particularly focused on microplastics, and they did the validation tests with passive particles in lines 352-354 as "One of the analyzed studies, Pilechi et al. (2022), did a comparative analysis of advection schemes to resolve the advection term in Equations 6 and 7 using a case study of microplastic transport in the Saint John River Estuary." and in line 356-357 as "Among these, the RK4 method emerged as the most accurate advection scheme when validated with the analytical test cases with passive particles,"

Line 369: The manuscript has a lot of unfamiliar acronyms, and I often have had to scroll back to figure out what they are. Suggest reducing the use of acronyms and instead use the words when possible – e.g., "random walk model" doesn't take up much ink on the page, and "RWM" is not a common term in the literature.

**We minimised the use of acronyms as suggested.**

Line 374: This is another example where the text stops at reporting what the previous authors

did rather than explaining why it matters or how it is relevant to the best practices for modelling plastics. There are lots of random walk models out there, and reviews of that topic that synthesize pros and cons. Here it would be helpful to distill how that choice influences outcomes for modeling plastics.

We agree with the reviewer's point. In this section, our intention was not to compare different random walk models but rather to highlight the novelty of this topic within the context of microplastic simulations in estuaries. Based on previous studies, we found that the diffusion calculation using Equation 10 is particularly appropriate when accounting for spatio-temporal variations in diffusivity, an approach applicable to both sediments and microplastics. While this point was already emphasized in our Recommendations section, we have also anticipated it in lines 372-374 as "This formulation can be particularly appropriate in estuarine studies as these systems can be subject to strong spatio-temporal variability of vertical mixing (Simpson et al., 1990; Burchard, 2009), potentially influencing microplastic transport."

Line 382: Given the common sources of uncertainty in the different modeling approaches (e.g., plastic terminal velocity), it would be useful to address that as a cross-cutting topic, in parallel with the division between Lagrangian and Eulerian transport frameworks.

We agree that this is a valuable suggestion and in fact considered this option during the writing process. However, while some sources of uncertainty are shared across modelling frameworks, others are specific to particular approaches. For that reason, we chose to structure this section, intended to be more descriptive, by modelling approach, which we still believe is a reasonable choice, especially since common parameters are discussed later in Sections 3.2 and 4 to discuss the choices from different studies and the challenges and recommendations.

Line 393: maybe I missed these acronyms...are PE, PS, and PP types of plastic? Please clarify.

**We included the acronyms in the text at line 393**

Line 395: "and rising" typo

**Done**

Line 443: Is "bubble's oxygen" modelling the oxygen inside bubbles? It's an unfamiliar term.

**Modified**

Line 445: The continuum of particle sizes seems like another topic which is cross-cutting and would be common to Eulerian or Lagrangian approaches. Presumably, the assumption in the cases above is that there is a single plastic size class, but that does not have to be the case. Suggest making the size distribution a separate point of discussion and including this Population Balance approach in its assessment.

While all approaches can account for more than one particle size class, a key distinction between the Eulerian/Lagrangian frameworks and the PBE lies in their treatment of particle size/characteristics: the former uses discrete classes, whereas the latter employs a continuous distribution. We highlighted this point in line 426-429 as "While PBE approach is conceptually similar to Eulerian approaches by estimating the time and space evolution of tracer concentration through an advection-diffusion equation, it

employs a continuous distribution of particle size, in contrast to ENM or LNM, which use discrete particle classes."

Line 452:Should the "eta" in this sentence be "xi"? There doesn't appear to be an eta in equation (28).

**Modified**

Line 459: The transport equation (28) is very similar to the transport equation (7) for the ERM, which raises questions about why it is categorized separately. It would be helpful to clarify the advantages of PBE that make it more efficient than ERM, and what that depends on (e.g., number of size classes, discretization of the Eulerian model, time step limitations for particle settling).

As explained in the previous comment, this is highlighted in lines 426-429 and in lines 444-446 as "PBE model avoids the limitations of ENMs, which typically rely on fixed particle classes and may struggle to capture shifts in particle distributions due to spatial or temporal variations."

Line 461: The suggestion here is that PBE handles "all particle sizes". Is PBE discretized in multiple, yet still discrete, size classes or a continuous function of particle size? Seems like there is still a tradeoff in the number/size of particle classes being represented, which would also be the case in a Eulerian or Lagrangian approach representing more than one particle size.

Same as explained in the previous comment, this is highlighted in lines 426-429 and lines 444-446

Line 473: Even if there are few examples, it would be useful to assess the PBE approach rather than just stating that it exists.

We agree with the reviewer that a more detailed assessment of the PBE approach would be valuable; however, to the best of our knowledge, the study we included is currently the only published application of this method in the context of microplastic modelling. Given that it is a very recent contribution, we focus on discussing its key aspects and highlighting it as a promising and novel direction.

Line 476: These assessments of advantages and limitations would fit better following the description of the approach, while the subject is still fresh in readers' minds. The description of each approach could be significantly shortened, and the pros and cons presented here could form the main message of that combined section.

Placing the advantages and limitations immediately after the description of each approach is also a good choice, which we also considered during the writing process. However, we prefer to highlight the comparative strengths and weaknesses of each approach by discussing them together in a dedicated section. This comparative format allows for a clearer understanding of how the methods differ in terms of their suitability and limitations across various scenarios.

Line 507: The report here is that Lagrangian models rarely incorporate key processes related to the ambient conditions such as erosion and deposition or buoyancy effects. However, it seems straightforward to incorporate these factors based on information from the circulation model (salinity/temperature, bottom stress). Is it a choice that previous applications have not done so or is

there some basic technological limitation? The following sentence says that it is easy to take into account these processes, referencing Jalon-Rojas et al. (2024b). Please clarify.

In our review of existing studies using Lagrangian numerical models (LNMs) for estuarine microplastic transport, we found that most applications did not incorporate key parameterizations related to deposition–resuspension processes or variations in water density (e.g., due to salinity or temperature). As the reviewer correctly noted, this omission is not due to a technological limitation of LPTMs themselves—these processes can indeed be incorporated, which we pointed out. Rather, we intended to highlight that, despite their importance in estuarine systems, these processes have rarely been considered in past studies using LPTMs.

Line 517: It is not obvious how initial conditions influence diffusion, either imposed by the turbulent diffusivities or due to numerical diffusion from advection. Please explain.

**We agree that this sentence can be confusing and removed it.**

Line 543: The overview framework described in this paragraph is clearly written and would be useful earlier in the manuscript, before the details on the different approaches to the transport modelling.

**As suggested by the reviewer, we moved this part into Section.2, lines 98-103**

Line 578: Perhaps it would be more succinct to say that the release strategy depends on the questions being addressed by the model, e.g., depending on the sources of plastic, forecast vs hindcast, environmental conditions influencing transport.

As per the reviewer's comment, we included the details about the choice of release strategies in lines 562-565 as "In summary, our review indicates that there is no general rule for defining release points, as this depends on each application, the actual sources of microplastics, the simulation type (forward or backwards), and local environmental conditions. It seems challenging to account for all the potential sources present in an estuarine environment, probably due to the lack of data and computational time limitations.

Line 603: Can convergence testing address questions on the number/concentration of particles to release, like what is done for evaluating model grid resolution or time stepping?

We provided details on the sensitivity tests involving particle concentration in the recommendations sections after providing the issue.

Line 608: For time stepping, it seems like the Courant condition relating grid size and advection length scale is an important consideration. Or perhaps that can be relaxed for Lagrangian models where length scales for velocity gradients are much larger than the grid size?

We described the Courant condition in the recommendation section along with the challenge.

Line 615: As above, it would be easier for readers to keep track of the issues and tradeoffs if the review sections and recommendations/analysis on each topic are presented together.

We have structured each challenge followed by its corresponding recommendation

in Section 4 of the manuscript. We aimed to present an integrated overview that addresses the challenges by considering all approaches together.

Line 616: It seems like the main message of this section (3.1) is that choices on release approach and simulation period depend on the study goals. In the interest of streamlining the text, that could be said succinctly rather than listing numerous examples that don't provide much additional insight but take up several pages of text.

While we agree that the main message of this Section is indeed that choices of the tracking parameters are closely tied to the study objectives, we believe it is important in the context of a review paper to provide a comprehensive overview of how these parameters have been handled in previous studies. Including these examples allows us to highlight the diversity of approaches used in the literature and helps readers understand the rationale behind different modelling decisions. We have aimed to balance clarity with completeness and believe this level of detail adds value for researchers looking to compare methodologies or design their studies.

Line 637: Is there anything about the specification of these mixing parameters that have considerations for plastics, rather than the general question of mixing (e.g., for salinity, sediment, or biota)? If so, that should be the focus of this section. Otherwise, the topic of mixing in estuaries is broad and complex, and can be referenced to other work.

The reviewer is right. The key point here is to emphasize that multiple options exist for the choice of mixing parameters, and we aim to provide an overview of the choices made in previous microplastic studies, along with the rationale behind them to support a later discussion of the associated challenges and recommendations. To clarify our intention, we added the lines 645-647 as "From the above discussions, it is clear that different studies have adopted various diffusivity coefficient values. The challenges associated with selecting appropriate mixing parameters have been further discussed in Section 4, along with the recommendations."

Line 754: This paragraph has broad generalities but little new content, and could be removed.

We agree with the reviewer that these are general recommendations; however, to our knowledge, no other review paper provides guidance specifically on modelling microplastic transport in estuaries, and we believe that such recommendations can be useful for researchers entering this field.

Line 778: As with the previous comment, the value added by this paragraph is limited because its content has been covered already above. Suggest moving the recommendations closer to where these issues are described earlier in this text and removing this rehash.

**We already responded regarding the organisation of the recommendation section**

Line 816: As above, move the recommendations closer to the discussion of the literature on particle characteristics.

As noted above, we have presented the challenges and corresponding recommendations in a dedicated final section, as we found this structure to be a more effective and coherent way to convey the key insights and guidance rather than including it in the discussion about each approach. Line 858: Missing "be" after "can"

Done